# Atomic resolution dynamics of cohesive interactions in phase-separated Nup98 FG domains

Eszter E. Najbauer[1,3], Sheung Chun Ng [2,3], Christian Griesinger [1], Dirk Görlich [2,4✉] & Loren B. Andreas [1,4✉]

Cohesive FG domains assemble into a condensed phase forming the selective permeability barrier of nuclear pore complexes. Nanoscopic insight into fundamental cohesive interactions has long been hampered by the sequence heterogeneity of native FG domains. We overcome this challenge by utilizing an engineered perfectly repetitive sequence and a combination of solution and magic angle spinning NMR spectroscopy. We map the dynamics of cohesive interactions in both phase-separated and soluble states at atomic resolution using TROSY for rotational correlation time (TRACT) measurements. We find that FG repeats exhibit nanosecond-range rotational correlation times and remain disordered in both states, although FRAP measurements show slow translation of phase-separated FG domains. NOESY measurements enable the direct detection of contacts involved in cohesive interactions. Finally, increasing salt concentration and temperature enhance phase separation and decrease local mobility of FG repeats. This lower critical solution temperature (LCST) behaviour indicates that cohesive interactions are driven by entropy.

[1] Department of NMR-based Structural Biology, Max Planck Institute for Multidisciplinary Sciences, Göttingen, Germany. [2] Department of Cellular Logistics, Max Planck Institute for Multidisciplinary Sciences, Göttingen, Germany. [3] These authors contributed equally: Eszter E. Najbauer, Sheung Chun Ng. [4] These authors jointly supervised this work: Dirk Görlich, Loren B. Andreas. ✉email: goerlich@mpinat.mpg.de; land@mpinat.mpg.de

Nuclear pore complexes (NPCs) are equipped with a permeability barrier, which controls nucleocytoplasmic exchange and consequently the macromolecular compositions of the nuclear and cytoplasmic compartments[1,2]. This barrier must meet apparently contradictory requirements. On the one hand, it acts as a sieve and suppresses the passage of "inert material" as small as ~5 nm in diameter[3], which prevents an uncontrolled intermixing of nuclear and cytoplasmic contents. At the same time, it allows facilitated passage of nuclear transport receptors (NTRs) that shuttle cargos across the barrier and against concentration gradients[1]. The transported cargoes can be as large as newly assembled 60S ribosomal subunits with a 25 nm diameter[4,5]. Facilitated transport rates can reach ~1000 translocation events or a mass flow of 100 MDa per NPC per second[6], and NPC passage of a single molecule is usually completed within milliseconds to tens of milliseconds[7–9], even for ribosomal subunits[10]. In fact, passage through the NPC barrier can occur so fast that it appears limited only by the diffusion through the bulk buffer and the barrier material behaves like a perfect sink for NTRs[6,11].

FG repeat domains[12,13] are intrinsically disordered protein (IDP) regions anchored to the NPC scaffold[14–17], containing Phe-Gly (FG) motifs. They are key to transport selectivity[13,18–24]. Not only do they bind to NTRs during facilitated translocation[18,25] but also engage in cohesive intra- and inter-molecular inter-repeat interactions to form the actual permeability barrier[11,19,20,23,26,27]. Some cohesive FG repeats were also found to bind to purified constituents of the NPC scaffold[20,28], suggesting that the FG repeats could seal the barrier towards the walls of the pore via cohesive interactions.

Spontaneous phase separation of purified cohesive FG domains in an aqueous solvent has been observed before[26,27,29], where two coexisting phases are formed: the "FG phase", which is a hydrogel-like phase enriched in the FG domain (local concentration of FG domain ~200 mg/ml[26]) with properties defined by the macromolecular assembly of FG domain molecules, and an aqueous phase where the FG domain molecules remain rather dilute (concentration < 0.5 mg/ml) and solvated by the aqueous solvent.

The FG phase mirrors the transport selectivity of NPCs very well[26,27,29] by allowing efficient entry of NTRs and NTR·cargo complexes to partition coefficients of ~1000, while excluding inert macromolecules (lacking favourable interactions with FG motifs) to partition coefficients of <0.1. The latter sieving effects require the high local FG domain concentration (~200 mg/ml) in the FG phase mentioned above[11,26]. This is in line with the observations that cohesive FG interactions are fundamentally required for NPC transport selectivity. Following entry, cargo·NTR complexes can traverse the FG phase, and cargoes can be rapidly released back into the aqueous phase[11,27,30]. This suggests that the interactions within the FG phase are dynamic and reversible, even if the FG domains within the FG phase show little to no detectable translational diffusion[26,27].

Yeast and vertebrate NPCs contain ~10 different FG domains, which differ widely in the prevalent FG motifs (e.g., GLFG, FSFG, or SLFG), FG motif density, and composition of the inter-FG spacer[26,31]. Earlier reconstitution experiments[23] suggested that the Nup98 FG domain[32] is the most critical one for barrier formation. It is enriched in GLFG motifs, has the highest FG-density (1 motif per ~12 residues), and is extremely depleted of charged residues – consistent with the view that FG domains experience water as a poor solvent[26].

The differences in the sequence and composition of FG domains can impact their biophysical properties. For example, the cohesive fungal FG domains are typically NQ-rich. Magic angle spinning (MAS) NMR analysis of an FG phase assembled from the *S. cerevisiae* Nsp1 FG domain identified strong cross-polarization (CP) signals for amyloid-like, rigid, inter-chain β-sheets, which originate from the NQ-rich N-terminal subdomain[33]. In addition, the FG domains from the *S. cerevisiae* Nup98 homologues Nup100 or Nup116 form phases that stain strongly with the amyloid-marker thioflavin-T. In contrast, less NQ-rich FG phases, e.g., *Tetrahymena thermophila* MacNup98A and *Xenopus laevis* Nup98 FG domains are essentially thioflavin-negative[26], and the latter also gives much weaker CP signals for inter-chain β-sheets[31]. In fact, the CP signal is completely lost when *Xenopus laevis* Nup98 FG domains receive their typical O-GlcNAc modifications[31]. This suggests that there may be different ways of assembling barrier elements but that amyloid-like structures are not fundamental for NPC function. In any case, the key role of FG motifs in cohesive interactions is conserved across species[19,20,23,26,33,34].

How the NPC barrier combines high transport capacity with selectivity is not fully understood. These permeability properties may be explained if cohesive interactions between FG repeats form transient meshes that are smaller than the passive size exclusion limit, while the binding of NTRs to FG motifs transiently opens those meshes, allowing NTRs to 'melt' through/partition into the barrier[6]. The surface properties of NTRs appear to be optimized for partitioning into the FG phase[29]. This hypothesis provides an explanation for the selectivity of nuclear pores based on the partition coefficient between the FG phase and the cytoplasm/nucleoplasm as well as an explanation for the fast transport rates of NTRs based on the local mobility of the barrier and weak transient interactions between FG motifs and NTRs[6,11,35,36]. While some features of the cohesive interactions have been well-defined by mutagenesis experiments[19,20,23,33,34], structural insight into the cohesive interactions is still missing but essential for obtaining a coherent picture.

From another perspective, the selective barrier in NPCs is an early example of biomolecular condensates, amongst meanwhile many well-characterized systems[37–41]. These condensates are assembled by phase separation of cellular proteins (and sometimes also nucleic acids) up to a protein concentration of 200–400 mg/ml and some of them exist as membraneless organelles. They are implicated in a wide range of biological events and also pathology[42,43]. Often, proteins involved in phase separation contain long (>100 residues) intrinsically disordered regions with low complexity sequences and are able to engage in multivalent low-affinity contacts[44,45]. The underlying interactions that drive phase separation are diverse for different condensates[46,47]. Many of the well-studied phase-separating IDPs are enriched in both charged and aromatic residues, which were proposed to be involved in electrostatic[39,48,49] and/or cation-π[38,40,50,51]/π–π[52] interactions. On the other hand, there are also examples deficient in both charged and aromatic residues, where hydrophobic effects have been implicated[53,54]. Nup98 FG domains are another extreme in the spectrum: they are highly enriched in aromatic residues (however, exclusively Phe) but deficient in charged residues.

NMR spectroscopy is often the method of choice for obtaining atomic-resolution insights into the dynamics and interactions of IDPs[36,55–57], and has also recently been used to uncover the driving forces of phase separation[53,56,58–67]. However, in the case of Nup98 FG domains, the length of any given native homologue (~500–700 amino acids) and the ~40–50 imperfect repeats[26] within the protein sequence present an obstacle. These sequences remain technically challenging for high-resolution NMR, due to the substantial overlap of resonances[57] and the presence of many similar sub-sequences. Since the overlap of resonances cannot be simply removed, we took the opposite route and utilized a perfect repeat (prf.GLFG$_{52\times12}$) consisting of 52 connected 12 amino-acid peptides GGLFGGNTQPAT, which simplifies the system to its essentials[27]. This allows residue-specific measurements using a combination of solution and MAS NMR spectroscopy, and

provides direct insight into the dynamics of cohesive interactions fundamental to NPC-barrier function.

## Results and discussion

**Phase separation of Nup98 FG domains exhibit entropy-driven LCST behaviour.** The engineered perfectly repetitive sequence, prf.GLFG$_{52 \times 12}$, captures several evolutionarily conserved features of native Nup98 FG domains. These include the overall amino-acid composition, the lack of charged residues, and the number and density of FG motifs. We previously showed that prf.GLFG$_{52 \times 12}$ phase separates like native Nup98 FG domains, including its parental FG domain, the wild-type *Tetrahymena thermophila* MacNup98A (Mac98A) FG domain (666 residues) (for sequences see Supplementary Note 1) and that the assembled phases showed similar permeation selectivity[27]. Both the prf.GLFG$_{52 \times 12}$ and Mac98A FG phases are simple systems that, unlike their animal orthologues, do not rely on obligatory O-GlcNAc modification. In addition, they are both thioflavin-T-negative, suggesting that they do not form amyloid-like structures.

We previously used polyhistidine-tagged versions of the FG domains. For the purpose of performing high-resolution NMR measurements in this study, we made a construct of polyhistidine-SUMO-prf.GLFG$_{52 \times 12}$, such that the unwanted histidine-tag could be removed by protease digestion[68] during purification (Methods). As shown in Fig. 1a, a 20 μM (~1.2 mg/ml) solution of the histidine-tag-free prf.GLFG$_{52 \times 12}$ self-assembled into an FG phase, in the form of scattered particles ("FG particles") at 21 °C. We found the FG domain concentration within the assembled FG phase to be ~4.5 mM (260 mg/ml), which corresponds to 230 mM of FG motifs (52 motifs per FG domain) (Supplementary Fig. 1). Thus, phase separation concentrated the protein more than 200-fold. The assembled FG phase showed essentially the same permeation selectivity as Mac98A FG phases: both FG phases excluded the inert probe mCherry protein (~30 kDa), but at the same time allowed strong partition (partition coefficients ~2000) of the nuclear transport receptor NTF2, which is similar in size to mCherry.

We noticed that the phase separation of both wild-type and perfectly repetitive FG domains was enhanced by increasing temperature (Fig. 1c–f and Supplementary Fig. 2: Mac98A). For example, 15 μM of prf.GLFG$_{52 \times 12}$ remained soluble at 4 °C, but phase separated readily at 37 °C and the process was reversible by changing the temperature (Fig. 1c). These observations led us to investigate the temperature dependence of phase separation systematically. Phase separation occurs when the protein concentration reaches a threshold (the "critical" or "saturation" concentration), which is also equal to the concentration in the aqueous phase after phase separation has occurred. We found that the critical concentration for phase separation decreased sharply from ~90 μM at 7 °C to ~1 μM at 37 °C (Fig. 1d).

Next, using comprehensive dynamic light scattering (DLS) measurements (Fig. 1e, f) we obtained a phase diagram (phase transition temperature against composition) for prf.GLFG$_{52 \times 12}$. Interestingly, we observed a concave curve, indicating a lower critical solution temperature (LCST) phase behaviour[69,70]. LCST-type phase separations are often driven by hydrophobic interactions i.e., unfavourable entropy of mixing a polymer and the solvent, which increases with increasing temperature. In this case, the water content of LCST-type condensed phases is predicted to be dependent on temperature: higher temperature increases the influence of entropy and since water molecules are restricted in their motion next to a hydrophobic group, water leaves the polymer (known as 'synersis'). Indeed, we observed this phenomenon for the FG phase by concentrating the gel-like phase and filling up a glass tube by centrifugation (Fig. 1g). We found that the

macroscopic FG phase shrank readily by ~50% in volume when the temperature was increased from 25 °C to 65 °C, indicating that water left the FG phase at higher temperature. Furthermore, we also found that the transition temperature decreased with increasing salt concentration (Fig. 1f), confirming the strong role of hydrophobic interactions. Accordingly, low salt concentrations disfavour cohesive interactions. From the above, conditions (temperature/ [NaCl]/ [FG domain]) for phase separation of prf.GLFG$_{52 \times 12}$, as well as conditions where prf.GLFG$_{52 \times 12}$ remains soluble ($c < c_{critical}$, and thus only the aqueous phase exists) can be derived from the phase diagram.

**Phase-separated Mac98A FG domain and prf.GLFG$_{52 \times 12}$ show low translational but high rotational mobility.** We analysed the microscopic dynamics of FG phases by fluorescence recovery after photobleaching (FRAP) experiments (Fig. 2a), which reveal the translational mobility of the whole molecules by correlating the diffusive recovery rate of a bleached region to a diffusion coefficient. To this end, we coupled the FG domains with the most weakly "FG-philic" fluorescent dye identified, Atto488[27], incorporated them into assembled FG phases, and performed photobleaching on the FG phases. The FG phase assembled from wild-type Mac98A FG domain showed essentially no recovery after bleaching[26] even after an hour, meaning that phase-separated FG domains are translationally rather immobile. prf.GLFG$_{52 \times 12}$, however, showed mobility to an estimated (translational) diffusion coefficient of ~0.06 μm²/s, however, this is still >600 times slower than that of its soluble state (~40 μm²/s, from DLS measurement). We previously showed that this difference (when compared to the Mac98A FG domain) is due to the lack of the 44-residue GLEBS domain (binding site for the mRNA export mediator Gle2p[71,72]) in prf.GLFG$_{52 \times 12}$[27], which also contributes to cohesive interactions, and suggested that the overall strength of cohesion could affect the translational mobility of molecules and viscosity of the FG phase material.

Next, we investigated the rotational diffusion of individual amino-acid residues, which is fundamental for understanding the permeability behaviour of FG phases. Since in NPCs the ends of FG domains are anchored to the NPC scaffold, and the FG domains are thus translationally immobile, the permeability of the barrier must depend on local dynamics. Therefore, we analysed the FG phases by NMR spectroscopy, measuring rotational dynamics of the FG domains[55,56]. For this purpose, we expressed and purified isotope-labelled prf.GLFG$_{52 \times 12}$ and the parental wild-type Mac98A FG domain from *E.coli* cultured in a modified M9 medium (Supplementary Table 2), as described in Methods.

We found that solution NMR measurements of the FG phases of both the Mac98A FG domain and prf.GLFG$_{52 \times 12}$ were not possible due to the severe line broadening caused by the magnetic susceptibility inhomogeneities in the viscous FG phases. Instead, we analysed the samples by MAS NMR spectroscopy[61,64,73], in which the samples were spun around an axis tilted at an angle to the magnetic field such that line broadening stemming from macroscopic magnetic susceptibility inhomogeneity was eliminated. We formed $^{13}C,^{15}N$-labelled FG phases inside MAS rotors and analysed them by 1D spectroscopy at low (20 kHz) MAS frequency and low-power proton decoupling. Although the aqueous phase coexists with the FG phase in the same rotor, signals stemming from the former are negligible due to the large difference in both protein concentration and volume of the phases.

To assess the local mobility of the peptide chain, we acquired direct excitation of $^{13}C$ and $^{1}H$-$^{13}C$ cross-polarization (CP) spectra (Fig. 2b). While the former shows all $^{13}C$ atoms, in the latter, signals

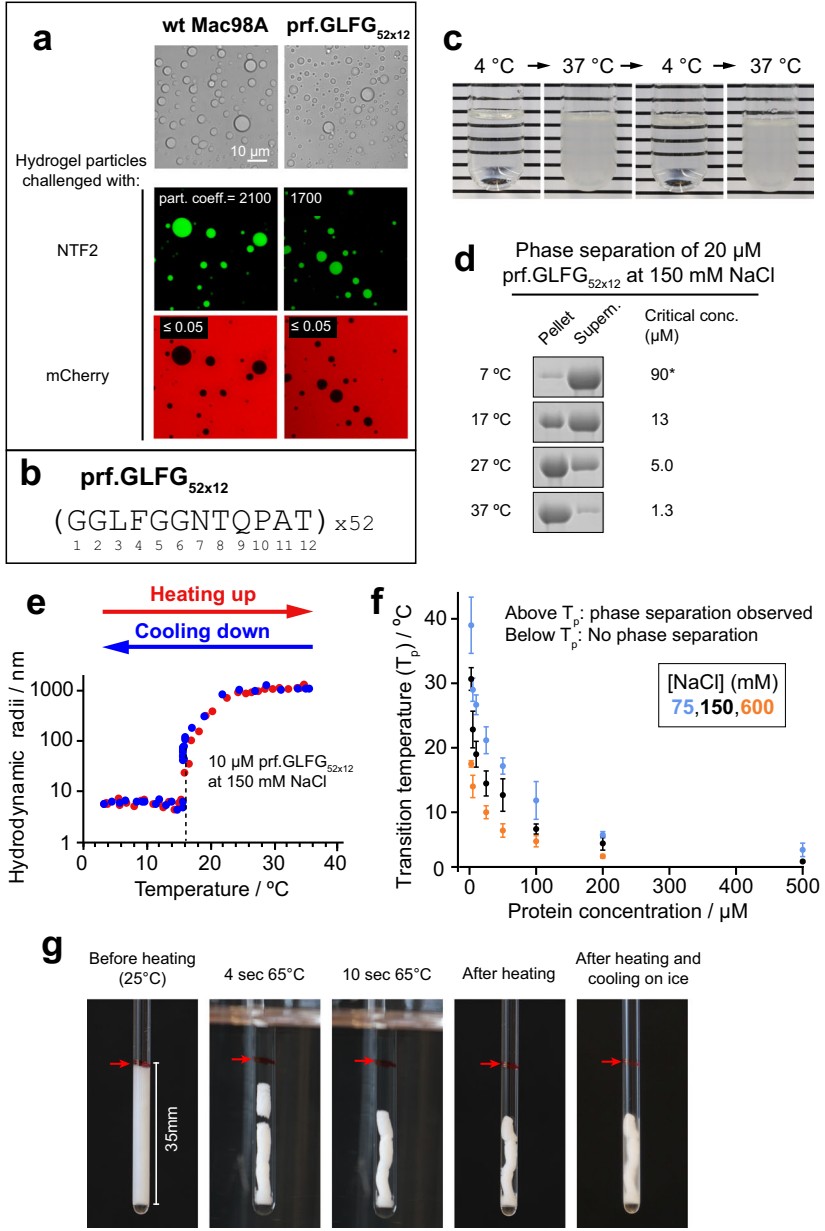

**Fig. 1 Phase separation of Nup98 FG domains shows a lower critical solution temperature (LCST) behaviour. a** Both the wild-type Mac98A FG domain and an engineered perfectly repetitive sequence, prf.GLFG$_{52\times12}$, phase separate to near-spherical particles following purification and dilution out of 2 M guanidine hydrochloride with an aqueous buffer (Methods), as shown by microscopy (top: phase-contrast microscopy images, below: confocal laser scanning microscopy images). Both FG phases exhibit NPC-like transport selectivity: they allow the partition of a nuclear transport receptor NTF2 (labelled with Alexa Fluor 488, green) to a partition coefficient of ~2000, but exclude the inert probe mCherry protein (partition coefficient < 0.05) (red). **b** Sequence of prf.GLFG$_{52\times12}$ containing 52 perfect repeats of a 12-mer peptide. **c** A dilution of prf.GLFG$_{52\times12}$ (15 µM at 400 mM NaCl) at 4 °C does not display phase separation, however when warmed up to 37 °C, it turns turbid readily, indicating phase separation. The process is reversible by cooling back to 4 °C. **d** 20 µM dilutions of prf.GLFG$_{52\times12}$ prepared at room temperature and centrifuged at different temperatures. SDS samples of the obtained pellets (FG phase) and supernatants (aqueous phase) are shown on SDS-PAGE, followed by Coomassie blue staining. Critical concentrations for individual conditions are taken as the concentrations of the supernatants. Full scans of gels with molecular weight markers are provided in the Source Data file. *At 7 °C, 100 µM of prf.GLFG$_{52\times12}$ is required to determine the critical concentration. **e** A dilution of prf.GLFG$_{52\times12}$ analysed by Dynamic Light Scattering (DLS), with the temperature increased from 2 to 40 °C (red). A sharp increase in hydrodynamic radii (from ~5 nm, expected for monomers[26,34], to final radii of ~1000 nm, expected for the multimeric FG particles) indicates phase separation at a transition temperature ($T_p$) ~16 °C. As in **c**, the phase separation is reversible by cooling (blue). **f** DLS analysis repeated with different concentrations of prf.GLFG$_{52\times12}$ and NaCl to determine $T_p$, which is plotted against prf.GLFG$_{52\times12}$ concentration. Data are plotted as mean values ± standard deviations (S.D.s) of three replicates. **g** Photographs of 30 mg of phase-separated prf.GLFG$_{52\times12}$ after centrifugation into a volume of ~150 µl at 25 °C, heated to 65 °C for 1 min, then cooled down on ice for 15 min. The red arrow marks the boundary between the condensed and the aqueous phase before heating.

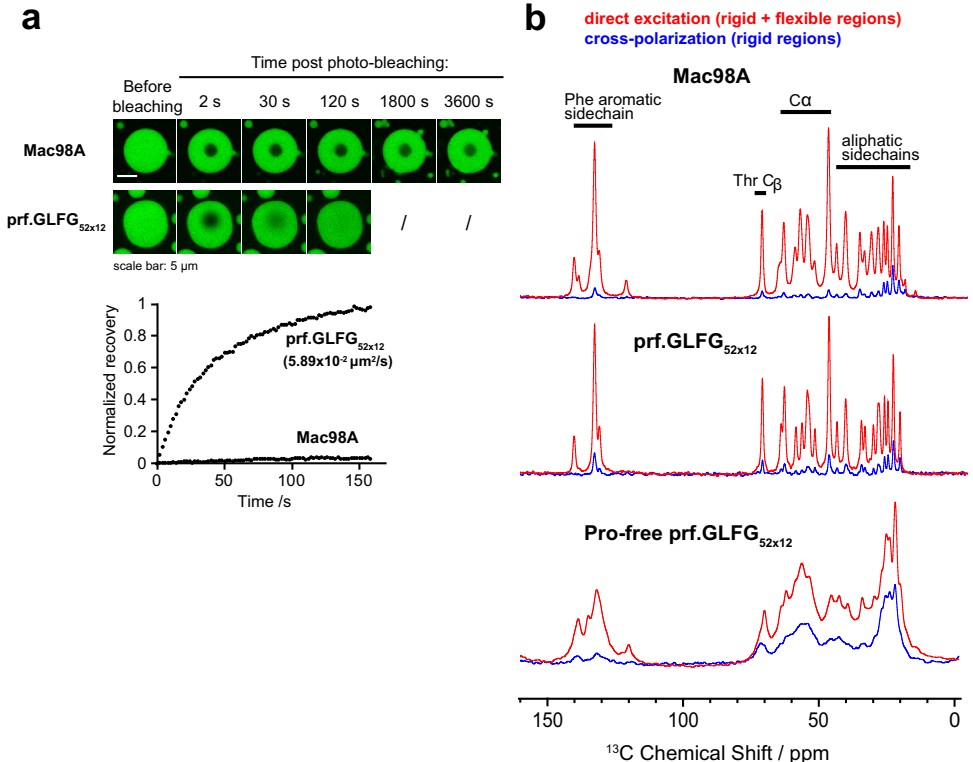

**Fig. 2 Phase-separated Nup98 FG domains show different behaviours at different scales. a** Recovery after bleaching of FG phases assembled from Mac98A FG domain or prf.GLFG$_{52 \times 12}$ spiked with 2.5% Atto488-coupled FG domains (of the same species). Bleached area: manually defined to be circular with a diameter of ~3 µm. Fluorescence recovery over time is shown (recovery is normalized to 1 for a complete recovery). For prf.GLFG$_{52 \times 12}$ the translational diffusion coefficient is shown, as derived from fitting the dataset to a theoretical exponential recovery curve (Methods). The Mac98A FG domain is essentially immobile. Although a slight increase of signal inside the bleached area was observed e.g., at 1800 s and 3600 s, this is likely due to self-recovery of the bleached fluorophore molecules but not due to diffusive recovery, as a change of postbleach profile[91] is not observed. **b** 1D $^{13}$C spectra of FG phases assembled from Mac98A FG domain (top), prf.GLFG$_{52 \times 12}$ (middle), and a thioflavin-T-positive, proline-free variant of prf.GLFG$_{52 \times 12}$: (GGLFGGATNSQT)$_{52}$ (bottom), obtained by MAS NMR (at 36 °C, 20 kHz MAS). In red, the signal from direct excitation (showing all $^{13}$C atoms), in blue, the cross-polarization (CP) spectrum (selective towards rigid parts of the samples). All spectra were obtained by averaging 1024 scans.

selectively stem from rigid parts of a given sample. The reason for this is that in cross-polarization, magnetization is transferred from $^{1}$H to $^{13}$C atoms through strong, orientation-dependent dipolar interactions. If the peptide chain exhibits fast (ns timescale) isotropic motion, these interactions are averaged, and magnetization transfer can only take place through much weaker through-bond scalar couplings[74], resulting in a much weaker CP signal. Interestingly, we found that both the wild-type Mac98A FG domain as well as prf.GLFG$_{52 \times 12}$ exhibit low CP intensity throughout the sequence (direct excitation $^{13}$C signals are over 10-fold stronger than the CP signals), indicating that both sequences lack regions with low mobility. Moreover, both exhibit similar liquid-like spectral characteristics, e.g., narrow ~70–80 Hz linewidths even at low MAS frequency and low-power decoupling. These indicate high local rotational mobility of both FG domains in FG phases on a nanosecond timescale despite their low translational mobility inferred from the FRAP experiments.

This finding has important implications, namely that the FG phases show a non-Brownian behaviour, with uncoupled translational and local rotational motion. Even though a molecule of FG domain is translationally rather immobile, the amino-acid residues can remain locally dynamic. This behaviour is similar to what was observed in various organic polymers (e.g., styrene-butadiene rubber[75]), where cross-links between polymer chains hinder translation, while locally the polymer chain displays solution-like mobility. Similarly, in these cases, the scale of observation determines whether liquid- or gel-like[76] properties prevail. We expect native FG domains in NPCs, where the ends of FG domains are anchored to the wall of the NPC, to also exhibit such non-Brownian behaviour.

MAS NMR is also sensitive to amyloid-like backbone interactions, which reduce mobility and thus increase CP intensity[33,77]. In contrast to Mac98A and prf.GLFG$_{52 \times 12}$ for which amyloid-like backbone interactions were not detected[26,27], an alternative perfectly repetitive sequence engineered previously, which lacks proline ("Pro-free prf.GLFG$_{52 \times 12}$" in Fig. 2b), stains very strongly with the amyloid-marker thioflavin-T[27]. The FG phase assembled from this sequence exhibits broad lines and a much higher CP signal, confirming the presence of amyloid-like structures in this type of FG phase.

**Simplification of sequence improves spectral resolution and enables sequence-specific assignment.** Due to the liquid-like spectral characteristics of both Mac98A and prf.GLFG$_{52 \times 12}$ FG phases at ambient temperatures, we adapted multidimensional solution NMR pulse sequences for use in proton-detected MAS NMR probes to derive precise residue-specific information on local dynamics. This combination of MAS and solution NMR is especially advantageous for FG phases, as the use of MAS eliminates line broadening, while scalar coupling-based solution NMR pulse sequences exploit long transverse relaxation times stemming from the inherent mobility of the peptide chain, yielding optimal sensitivity.

We first assessed spectral resolution in 2-dimensional amide nitrogen ($^{15}$N) and proton (H$^N$) ($^{15}$N-$^1$H HSQC$^{SSMAS}$, Fig. 3a) correlation spectra, where different peaks reflect the different chemical environments experienced by the NH groups. Scalar couplings were used for magnetization transfer, as dipolar couplings are averaged out by the fast (ns timescale) isotropic motion of the peptide chain. Unsurprisingly, we found that the spectrum of the Mac98A FG domain suffered from severe signal overlap due to the >600 non-equivalent amide groups in the sequence[57]. The spectrum of prf.GLFG$_{52 \times 12}$, however, displayed an isolated peak for each of the residues within a repeat unit (except Pro, which does not have H$^N$), while showing a very similar overall spectral pattern with the strongest signals from the Mac98A FG domain; note the overlap of both spectra in Fig. 3a. This suggests that both sequences experience similar chemical environments and is characteristic of IDPs, for which the neighbouring residues largely dictate the amide chemical shifts.

We then focused on prf.GLFG$_{52 \times 12}$ for detailed measurements of dynamics and interactions. The narrow linewidths in the $^{15}$N-$^1$H HSQC$^{SSMAS}$ spectra ($^{15}$N: < 0.19 ppm, $^1$H: < 0.051 ppm), and the only slightly narrower linewidths in the TROSY spectrum ($^{15}$N: < 0.14, $^1$H: < 0.046 ppm) confirmed the inherent mobility of the peptide chain, while the narrow chemical shift dispersion of the proton resonances indicates intrinsic disorder of both sequences (wt Mac98A and prf.GLFG$_{52 \times 12}$) in the FG phases. To confirm this, we compared the $^{15}$N-$^1$H correlation spectra of prf.GLFG$_{52 \times 12}$ with spectra obtained under strong denaturing conditions (i.e. in the presence of high concentrations of guanidine hydrochloride or hexanediol, which impede phase separation[26]) (Supplementary Fig. 3). The similar proton chemical shift dispersion indicates intrinsic disorder under both conditions. This is consistent with other proteins that undergo phase separation in that the protein can remain disordered in both the aqueous phase and in biomolecular condensates[58,59,62,63,78].

Three-dimensional HNCA and HN(CO)CA spectra correlating N and H$^N$ with the Cα of the same and the neighbouring residue, respectively, were implemented as typical for solution samples, correlating the nuclei via scalar couplings. These two spectra were sufficient to unambiguously assign the 11 observed backbone amide peaks in the $^{15}$N-$^1$H HSQC$^{SSMAS}$ spectrum to the 11 non-proline residues of prf.GLFG$_{52 \times 12}$. This sequence assignment is the foundation for further residue-specific measurements, and also implies that the different repeats are indistinguishable by NMR (Fig. 3b and Supplementary Fig. 4).

The $^{13}$C-$^1$H correlation spectrum ($^{13}$C-$^1$H HSQC$^{SSMAS}$) of prf.GLFG$_{52 \times 12}$ (Fig. 3c) reflects the chemical environments experienced by the CH groups, including those in the Phe aromatic sidechain. Three peaks were observed in the aromatic region of the spectrum at 2:2:1 intensity, corresponding to the three non-equivalent CH groups in the aromatic sidechains, showing that the chemical shifts of the Phe residues are averaged faster than the μs-timescale. The narrow lines observed in the $^{13}$C-$^1$H HSQC$^{SSMAS}$ spectrum, even for the aromatic sidechains, further indicate that the motion of the peptide chain occurs on a nanosecond timescale, fast enough to isotropically average even strong orientation-dependent interactions such as dipolar couplings.

**Effect of phase separation, temperature and salt concentration on local dynamics of the peptide chain points to the importance of hydrophobic interactions in cohesive behaviour.** After analysing the FG domains in the FG phases, we were interested in comparing the NMR spectra of prf.GLFG$_{52 \times 12}$ in the FG phase and in the complementary aqueous phase (HSQC$^{SSMAS}$ and HSQC, respectively), which may show residues involved in phase separation. We had measured $^{15}$N-$^1$H and $^{13}$C-$^1$H correlation

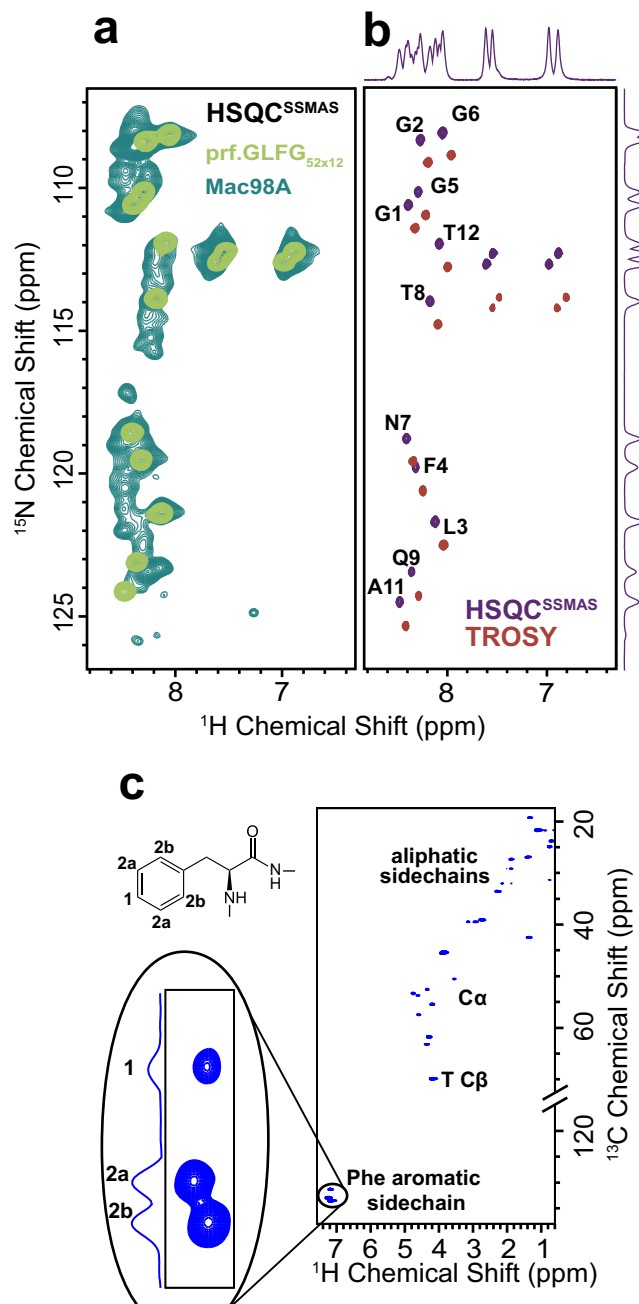

**Fig. 3 The engineered perfectly repetitive sequence, prf.GLFG$_{52 \times 12}$ improves the resolution of NMR measurements. a** Overlay of $^{15}$N-$^1$H correlation spectra ($^{15}$N-$^1$H HSQC$^{SSMAS}$) of phase-separated Mac98A FG domain (forest green) and prf.GLFG$_{52 \times 12}$ (light green), where each peak corresponds to an amide group with a distinct chemical environment. **b** $^{15}$N-$^1$H HSQC$^{SSMAS}$ of prf.GLFG$_{52 \times 12}$ (purple) and the TROSY spectrum (red). The peak positions are shifted by half the $^1$J($^{15}$N,$^1$H) coupling in both dimensions as expected. Assignments of the prf.GLFG$_{52 \times 12}$ are shown beside the peaks (also see Supplementary Fig. 4). Proline residues lack H$^N$, and thus are not detected. **c** $^{13}$C-$^1$H HSQC$^{SSMAS}$ spectrum of prf.GLFG$_{52 \times 12}$. The cross peaks corresponding to CH bonds of the Phe aromatic sidechain are enlarged and labelled with their relative intensities (2:2:1).

spectra for the former without any difficulty, as the protein concentration in the FG phase is in the millimolar range (Supplementary Fig. 1), providing strong signal intensities. For the latter, we prepared a sample in the soluble state based on

conditions established in Fig. 1f. We benefited from the fact that signals from the 52 repeats in prf.GLFG$_{52 \times 12}$ accumulate, thus we were able to detect signals from as low as ~5 μM of prf.GLFG$_{52 \times 12}$ by solution NMR. Supplementary Fig. 5a shows the overlay of $^{15}$N-$^1$H and $^{13}$C-$^1$H correlation spectra in both states. While the $^{15}$N-$^1$H correlation spectra overlap very well, we observed chemical shift perturbations larger than average for residues in and around the hydrophobic patch of the repeat, especially for F4 (Supplementary Fig. 5c). These perturbations are, however, hard to interpret, since amide protons—capable of hydrogen bonding—are sensitive to even a slight change in conditions. The $^{13}$C-$^1$H correlation spectra also overlap very well, with practically identical peak positions for all groups (Supplementary Fig. 5b). Slightly larger than average chemical shift perturbations can be observed, however, on the CH-groups of several hydrophobic/aliphatic side chains, (e.g. L3, F4, P10, A11) (Supplementary Fig. 5d).

To further investigate the origin of these chemical shift perturbations, we assessed local mobility by measuring rotational correlation times in both states of prf.GLFG$_{52 \times 12}$. For this purpose, we applied the [$^{15}$N, $^1$H]-TRACT pulse sequence, which allows the measurement of exact rotational correlation times ($\tau_c$) for Brownian particles by eliminating the effect of chemical exchange[79]. For the FG phase, an assumption of Brownian-like local dynamics, with an exponential rotational correlation function can still be used to assess qualitative changes in dynamics. Note that these qualitative changes could be interpreted in the context of different motional models, including anisotropic rotational diffusion, however, this would require extensive further relaxation measurements. We observed high, nanosecond timescale mobility in the soluble state (i.e., short $\tau_c$-values, Fig. 4). The profile of $\tau_c$-values is rather flat and all residues exhibit roughly the same value (~2 ns at 24 °C). In contrast, in the FG phase, not only is $\tau_c$ ~3-fold longer on average, but also the profile shows two distinct maxima. The maximum at the residues around P10 (Q9 and A11) can be explained by proline's fixed φ angle ($\varphi \approx -60°$), which limits the number of possible conformations in the five-membered pyrrolidine ring, and introduces rigidity into the protein backbone[80]. Proline also restricts the conformations available to the neighbouring residues, especially the preceding one[81], and this also appears in their increased $\tau_c$-values. The other maximum appears at residues F4 and L3. Phe residues were previously shown to be crucial for phase separation behaviour and thus both hydrophobic and π-π interactions of the aromatic rings likely contribute to cohesive interactions[19,20,82]. Such interactions are consistent with the increased $\tau_c$ of the peptide chain at this position.

Since all repeats are identical, FG repeats could cohesively interact with each other within the same molecule even in the soluble state, where intermolecular cohesive interactions are too weak to drive molecular assembly. Notably, with a ~5 nm hydrodynamic radius as determined by DLS measurements, the local concentration of FG motifs in the FG domain in the aqueous phase could also reach ~170 mM, similar to the concentrations of FG motifs in FG phase. Indeed, other cohesive FG repeats were previously shown to participate in intramolecular interactions[34,82]. To detect the effect of intramolecular cohesive interactions in the soluble state, we first measured rotational correlation times, $\tau_c$, for a fragment of the full-length protein sequence, the prf.GLFG$_{7 \times 12}$ peptide, which contains only seven of the perfect repeat units, thereby disfavouring the formation of intramolecular interactions by decreasing the probability of interacting with intramolecular cohesive regions. The nearly perfect overlap of the $^{15}$N-$^1$H HSQC spectra of the prf.GLFG$_{7 \times 12}$ fragments with the full-length protein eliminated the need for sequence assignment (Fig. 4a). Average $\tau_c$-values measured for both the prf.GLFG$_{52 \times 12}$ (aqueous state) and the prf.GLFG$_{7 \times 12}$ fragments

at 24 °C are significantly shorter than predicted by Stokes' law for spherical molecules (prf.GLFG$_{52 \times 12}$: 1.4 ns vs. 22.3 ns, prf.GLFG$_{7 \times 12}$: 0.7 ns vs. 4.0 ns, assuming a hydration layer of 3.2 Å)[83], showing that neither of the molecules tumbles as a whole. We found that prf.GLFG$_{7 \times 12}$ exhibited ~2-fold shorter $\tau_c$ than aqueous phase prf.GLFG$_{52 \times 12}$ (Fig. 4b). However, this difference could not only stem from a reduction of cohesive interactions, but in part also from the reduction of the overall correlation time due to the >7 times shorter peptide[84]. We thus took a complementary approach to disfavour the intramolecular cohesive interactions by lowering the salt concentration, suggested by what we found earlier in Fig. 1f. Indeed, we found that correlation times were reduced for both aqueous phase prf.GLFG$_{52 \times 12}$ and prf.GLFG$_{7 \times 12}$ in the absence of NaCl (Fig. 4c), suggesting that part of the changes in rotational dynamics resulted from changes in intramolecular cohesive interactions. Since the average $\tau_c$ for the prf.GLFG$_{7x12}$ is still shorter by 30% compared to that of prf.GLFG$_{52 \times 12}$ the overall size of the two peptides also contributes to the difference in rotational correlation time as predicted[84].

Given the observations of temperature-dependent phase separation, we also investigated the effect of temperature on local mobility by measuring rotational correlation times at 36 °C in both the aqueous phase and the FG phase. $\tau_c$ of the full-length protein in the aqueous state could not be determined at 36 °C due to the extremely low, ~1 μM critical concentration at this temperature. For prf.GLFG$_{7 \times 12}$, as expected, we observed that $\tau_c$-values dropped by half upon an increase of 12 °C, indicating increased mobility at higher temperatures. Strikingly, however, the peptide chain of the full-length prf.GLFG$_{52 \times 12}$ in the FG phase became less mobile; $\tau_c$-values increase almost twofold with the 12 °C increase in temperature (on average from 6 ns to 11 ns). The shape of the profile also changed, with L3 becoming less mobile relative to F4 at the higher temperature. These observations point to the role of hydrophobic interactions in the FG phase.

Moreover, consistent with Fig. 1f, we found that higher NaCl concentration also lowered critical concentrations and drove phase separation of prf.GLFG$_{52 \times 12}$ more to completion (Fig. 5a). Furthermore, we found that both microscopic and local dynamics of the FG phase of prf.GLFG$_{52 \times 12}$ are also dependent on salt concentration. FRAP experiments suggest >3-fold slower translational diffusion (Fig. 5b) upon an increase in NaCl concentration from 150 mM to 600 mM. We also measured residue-specific $\tau_c$ in the FG phase at different NaCl concentrations (Fig. 5c). The overall pattern of $\tau_c$-values with respect to residue number remains the same at each NaCl concentration, with maxima at L3-F4, and Q9-A11, and an increase in salt concentration results in an increased $\tau_c$ for each residue. The relative increase in $\tau_c$ is similar throughout the sequence, suggesting that not only the interacting hydrophobic residues, but the whole chain becomes less mobile at higher salt concentrations. The increase in phase separation propensity and the slowing of the dynamics of FG phase (due to stronger inter- and intramolecular interactions) at higher salt concentrations indicates once again the importance of hydrophobic interactions in cohesive interactions/ phase separation of the FG domains.

**NOESY reveals spatial proximities between aromatic-aliphatic and aliphatic-aliphatic hydrophobic groups.** In order to observe hydrophobic interactions in the FG phase, we recorded a $^{13}$C-filtered NOESY spectrum (Fig. 6), in which cross peaks are a direct indicator of spatial proximity between protons. In general, since the nuclear Overhauser effect (NOE) is proportional to $r^{-6}$, only protons closer than 5.5 Å are expected to give a NOESY cross peak. This means that protons within the same or in neighbouring residues will usually show cross peaks, as the length

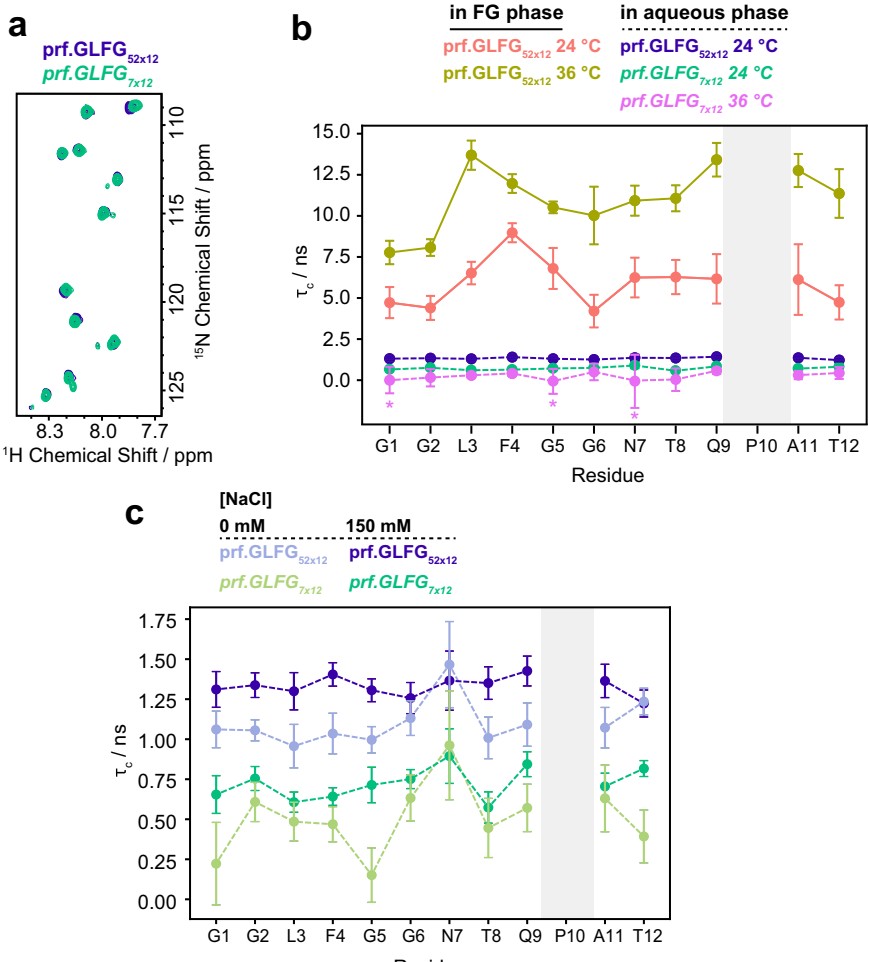

**Fig. 4 Effect of temperature, phase separation and salt on the residue-specific dynamics of prf.GLFG$_{52 \times 12}$. a** $^{15}$N-$^{1}$H HSQC spectra of the prf.GLFG$_{7 \times 12}$ peptide (green) and the full-length prf.GLFG$_{52 \times 12}$ protein (blue) in solution at 24 °C. **b** Residue-specific rotational correlation times ($\tau_c$) as determined by [$^{15}$N,$^{1}$H]-TRACT[79] for prf.GLFG$_{52 \times 12}$ in either FG phase or aqueous phase, and prf.GLFG$_{7 \times 12}$ at different temperatures in a 20 mM sodium phosphate buffer at pH 6.8 containing 150 mM NaCl. Rotational correlation times are plotted as calculated from the best fit values of $R_\alpha$ and $R_\beta$ from a single measurement. Error bars are shown as ±standard deviations (S.D.s). S.D.s were obtained from the exponential fits of the transverse relaxation rates $R_\alpha$ and $R_\beta$, and the Monte Carlo method ($N = 10000$) was then used to estimate the S.D.s of rotational correlation times. The profile of the full-length prf.GLFG$_{52 \times 12}$ at 36 °C (yellow-brown), and at 24 °C (peach) in the FG phase is shown with solid lines. Dashed lines indicate measurements in solution: full-length prf.GLFG$_{52 \times 12}$ in soluble state at 24 °C (blue), prf.GLFG$_{7 \times 12}$ at 24 °C (green), and prf.GLFG$_{7 \times 12}$ at 36 °C (pink). The dynamics of the full-length construct could not be determined in solution at 36 °C due to its extremely low critical concentration. At 36 °C, the exact correlation times of G1, G5, and N7 in the prf.GLFG$_{7 \times 12}$ peptide could not be determined, as the relaxation rates of α and β states of the amide $^{15}$N are similar within the range of errors. The $\tau_c$-values of these three residues are marked with an * and plotted as 0. **c** Aqueous phase full-length prf.GLFG$_{52 \times 12}$ and prf.GLFG$_{7 \times 12}$ in 20 mM sodium phosphate buffers at pH 6.8 containing either 0 mM or 150 mM NaCl and measured by TRACT at 24 °C.

of a residue is 3.4–4 Å[85–87]. These cross peaks do not necessarily indicate informative residue–residue interactions. Similarly, peaks appearing on the diagonal of the spectrum primarily stem from magnetization not transferred during the mixing period, which obscures contacts between equivalent groups, e.g., Phe-Phe interactions. However, after the assignment of all proton resonances of prf.GLFG$_{52 \times 12}$, we could identify several interesting contacts between non-neighbouring residues.

Although in our case quantitatively correlating NOE cross peak intensities to distance restraints is problematic with such a dynamic system, the observed cross peak pattern points to hydrophobic groups being in close proximity within the FG phase. Considering that some neighbouring protons show no correlations at all (e.g. N7 and F4 Hβ$_{1,2}$ and their neighbours), even cross peaks between neighbouring residues can be taken as an indication of increased proximity (e.g. L3 Hδ$_{1,2}$-F4 aromatic protons, in agreement with previously observed Leu-Phe contacts

in the Nsp1 FG phase[33]). Remarkably, we observed several cross peaks between protons from non-neighbouring residues (marked by arrows in Fig. 6), most prominently between the aromatic sidechain of F4 and the methyl groups of T8/T12, and also between purely aliphatic groups, such as T8/T12 methyl groups and the P10 β-methylene group. The relatively long mixing time of 100 ms has been used in investigations of other phase-separated systems as well[59,63,67]. As mentioned above, a limitation of the current method is that it does not allow for observing contacts between equivalent groups, therefore, proximity between Phe–Phe sidechains was ambiguous in our dataset, even though they are likely present in the system.

**Mutagenesis experiments suggest that sequence hydrophobicity determines phase separation behaviour.** Finally, observations from NMR studies guided us to the following series

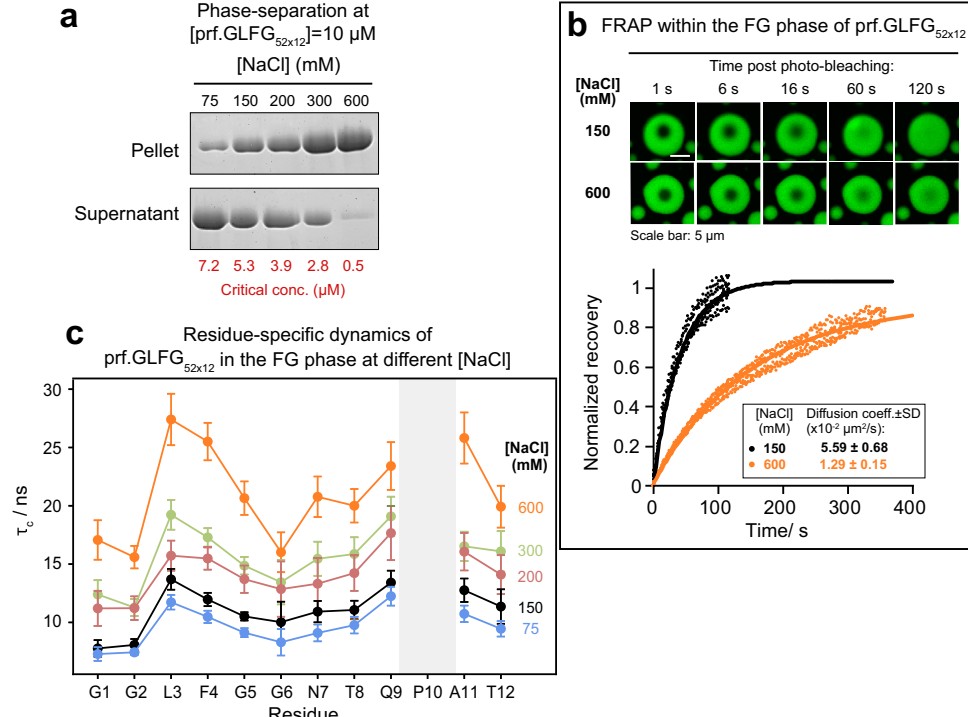

**Fig. 5 Effects of salt concentration on the properties of the FG phase. a** Analysis of phase separation by centrifugation as described in Fig. 1d with varying [NaCl] at 27 °C. Full scans of gels with molecular weight markers are provided in the Source Data file. **b** FG phase assembled from prf.GLFG$_{52 \times 12}$ containing 2.5% Atto488-coupled prf.GLFG$_{52 \times 12}$ molecules photobleached in 150 or 600 mM NaCl and the fluorescence recovery after photobleaching. *Top*: representative images showing the recovery of Atto488 signal (bleached area: manually defined to be circular with a diameter of ~3 μm). *Bottom*: Signal recovery normalized to 1 for a complete recovery plotted against time post bleaching. Fits (solid lines) to five replicates are shown. The translational diffusion coefficients (mean ± S.D.) derived from the fits are shown in the box. **c** Residue-specific rotational correlation times ($\tau_c$) of prf.GLFG$_{52 \times 12}$ in FG phase as determined from the [$^{15}$N,$^{1}$H]-TRACT experiment, shown as a function of [NaCl]. $\tau_c$-values are plotted as calculated from the best fit values of $R_\alpha$ and $R_\beta$ from a single measurement. Error bars are shown as ±S.D.s. S.D.s were obtained from the exponential fits of the transverse relaxation rates $R_\alpha$ and $R_\beta$, and the Monte Carlo method ($N = 10000$) was then used to estimate the S.D.s of $\tau_c$. The temperature was set to 36 °C for all measurements. Due to the absence of H$^N$, no TRACT data is available for Pro residues.

of mutagenesis experiments. As a basis for these, we chose the previously engineered FG domain variant 'GLFG$_{52 \times 12}$'[27], which contains 52 GLFG motifs connected by non-identical eight-residue spacers. This variant is as cohesive as the Mac98A FG domain. Unlike prf.GLFG$_{52 \times 12}$, whose amino-acid composition and locations are restricted by the repetitive nature, GLFG$_{52 \times 12}$ offers more flexibility in variant design (Fig. 7a and Supplementary Note 1). Here, GLFG$_{52 \times 12}$ showed a critical concentration of 0.1 μM under our assay condition (25 °C, [NaCl] = 150 mM). First, we mutated each Phe to Ala (GLAG$_{52 \times 12}$), as it had previously been shown that Phe residues are crucial for phase separation[19,20,23,33,34]. Indeed, we observed essentially a complete loss of phase separation behaviour, even up to 100 μM of GLAG$_{52 \times 12}$ (Supplementary Fig. 6). Since we had observed lower mobility not only for Phe but also for Leu residues, pointing to the importance of non-aromatic hydrophobic groups in phase separation, we designed a sequence with all Leu residues replaced by less hydrophobic Ala (GAFG$_{52 \times 12}$). Remarkably, this sequence was also rather non-cohesive and behaved similarly to GLAG$_{52 \times 12}$.

Native Nup98 FG domains have most Leu adjacent to Phe in the form of GLFG motifs or SLFG motifs, but Leu residues are also present in the spacers. We asked if the vicinity of Leu and Phe is required for phase separation and thus we swapped the positions of each Leu with a residue in the middle of the adjacent spacer in GLFG$_{52 \times 12}$. The corresponding variant, GXFG//L$_{52 \times 12}$ remained cohesive and exhibited only a slightly higher critical concentration (0.8 μM) than GLFG$_{52 \times 12}$. Interestingly, replacing

all Leu residues in the spacers with Val, which is only slightly less hydrophobic, already increased the critical concentration ten-fold to 8 μM (GXFG//V$_{52 \times 12}$), indicating that the system is highly sensitive to changes in hydrophobicity. These results indicate that not only Phe but also Leu residues are required for cohesive behaviour, even in the presence of all Phe[20,34].

Next, to prove our hypothesis, that it is the overall hydrophobicity of the sequence that determines the cohesive behaviour, we constructed a mutant (GLLG$_{52 \times 12}$), where all Phe were mutated to Leu. This variant showed no cohesive behaviour, its critical concentration being above 100 μM. However, this loss in cohesiveness could be caused by the fact that the Leu sidechain contains fewer hydrophobic carbons than that of phenylalanine. To compensate for the F→L substitution, we produced a Phe-to-double-Leu variant, GLLG//L$_{52 \times 12}$, where not only was the Phe substituted by Leu, but one more hydrophilic spacer residue was replaced by leucine. Remarkably, we found that this Phe-deficient GLLG//L$_{52 \times 12}$ variant showed phase separation at a submicro-molar critical concentration at our assay condition. The assembled 'leucine-rich phase' is in the form of spherical particles which exclude mCherry very well, showing a barrier-like property (Supplementary Fig. 7). This phenomenon further confirms the important role of Leu in the FG phase.

We also measured the intra-phase diffusion of GLFG$_{52 \times 12}$, GXFG//L$_{52 \times 12}$, and GAFG$_{52 \times 12}$ by labelling them with Atto488, incorporating them into the FG phase assembled from prf.GLFG$_{52 \times 12}$ and performing FRAP experiments (Fig. 7b, c). Both Atto488-labelled GLFG$_{52x12}$ and GXFG//L$_{52x12}$ partitioned

**Fig. 6 ¹³C-filtered ¹H-¹H NOESY spectrum of the FG phase of prf.GLFG$_{52 \times 12}$.** Cross peaks in this spectrum indicate spatial proximity between protons. Arrows mark cross peaks belonging to correlations between non-neighbouring residues and represent unambiguous spatial proximities between hydrophobic groups. Due to the large dynamic range of peaks, the contour levels were adjusted as follows: 1x (green peaks), 2x (red peaks) and 9.5x (dark blue peaks), the latter colour indicating the strongest signals. The applied mixing time was 100 ms and the measurement temperature was 31 °C. Dashed lines guide the eye to the assignments of chemical shifts; in case of overlap all possibilities are listed. The cross peaks indicating contacts involving the aromatic phenylalanine sidechain are boxed, and the skyline projection of the boxed region is shown next to the box.

well into the prf.GLFG$_{52 \times 12}$ FG phase (partition coeff. > 1000) and showed diffusion rates of ~0.05–0.07 µm²/s. These values are similar to the self-diffusion rate of prf.GLFG$_{52 \times 12}$ (Figs. 2a and 5b). However, GAFG$_{52 \times 12}$ not only partitioned very weakly into the prf.GLFG$_{52 \times 12}$ phase (partition coeff. = 4), but also showed >2 times faster diffusion compared to the above. GLLG//L$_{52 \times 12}$ partitioned readily into the prf.GLFG$_{52 \times 12}$ phase, but at a lower partition coefficient (110) compared to that of GLFG$_{52 \times 12}$ or GXFG//L$_{52 \times 12}$, while exhibiting diffusion behaviour similar to the Phe-containing sequences. Thus, we concluded that strength of cohesive interactions, as well as intra-phase dynamics are dependent on hydrophobicity contributed by both Phe and Leu.

The permeability barrier of NPCs has to fulfil requirements of both high transport capacity and high selectivity. Cohesive interactions are fundamental for the formation of such a barrier, concentrating and assembling the FG repeats into a condensed phase, while the permeation selectivity is observed because mobile species encounter the chemical properties of the FG phase, and beyond a certain size they must overcome the self-affinity, i.e., cohesion, of the FG repeats. To conclude this study, our approach allows the measurement of residue-specific local dynamics, through which we could begin to understand the dynamics of inter-FG repeat cohesive interactions within the FG phase. FG domains in the FG phase exhibit slow translational motion, but

fast local rotational motion. This apparent contradiction reflects the biological requirements of the NPC barrier, which needs to be kinetically stable to repel inert macromolecules but dynamic enough to allow fast facilitated passage by NTRs, as suggested before[6]. Although we do not directly measure on/off rates of cohesive FG repeat interactions, the consistent observation of high mobility on a nanosecond timescale and the promiscuous interactions of Phe sidechains suggest that the cohesive FG repeats interact with each other with high on/off rates without a conformational selection mechanism. This may have two consequences:[6] first, even weak NTR-FG repeat interactions could efficiently compete with the attraction between the FG repeats. Second, these interactions during translocation would require low activation energies as the entropic penalty is minimized. Notably, the binding without conformational selection may also extend to the interactions between FG repeats and NTRs[36]. Moreover, in contrast to mutagenesis approaches, we could also directly detect contacts involved in cohesive interactions.

An ultimate goal within the field is understanding all transport-relevant interactions within an FG phase and their dynamics, however, a number of issues remain unsolved. For example, although we showed that phase separation of the studied FG domains is driven mainly by entropy (hydrophobic interactions), the

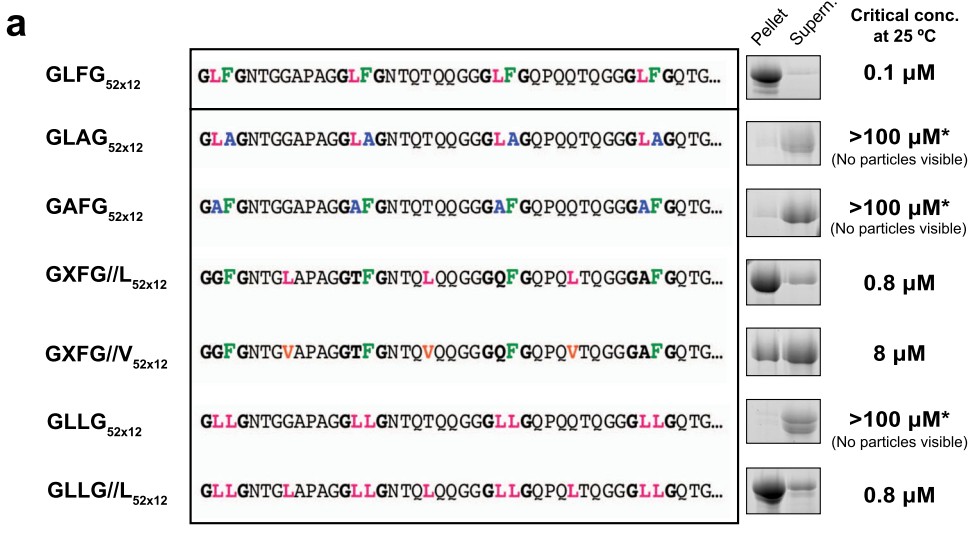

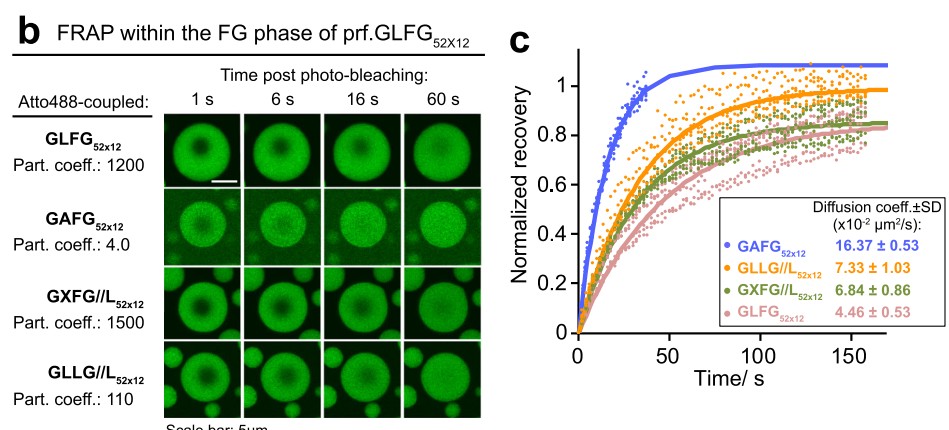

**Fig. 7 Influence of hydrophobic residues on the phase separation of FG domains. a** Critical concentrations for phase separation were determined on a series of GLFG$_{52 \times 12}$ variants. GLFG$_{52 \times 12}$ is an engineered FG domain variant, which contains 52 GLFG motifs flanked by non-identical sequences (spacers) of equal length. The spacers do not contain Phe and Leu. '//' indicates the swapping of a spacer residue with a residue from the GLFG motif. For a detailed description of the variants, see the main text. For space economy, the N-terminal sequences up to the fourth FG motif are shown, while the C-terminal sequences (~620 residues) follow the same design strategy (see Supplementary Note 1 for complete sequences). Phase separation of the variants were analysed as in Fig. 1d at 25 °C, at [FG domain] = 10 μM, [NaCl] = 150 mM. *The analyses for GAFG$_{52 \times 12}$, GLLG$_{52 \times 12}$ and GLAG$_{52 \times 12}$ were repeated at [FG domain] = 100 μM, and no significant phase separation was observed (Supplementary Fig. 6). Full scans of gels with molecular weight markers are provided in the Source Data file. **b** Images showing FG particles of unlabelled prf.GLFG$_{52 \times 12}$ premixed with 2.5, 15, 2.5 and 2.5% (by mole fraction) of Atto488-coupled GLFG$_{52 \times 12}$, GAFG$_{52 \times 12}$, GXFG//L$_{52 \times 12}$, and GLLG//L$_{52 \times 12}$, respectively. Apparent partition coefficients of the Atto488-coupled species into the assembled FG particles are indicated on the left as quantified by measuring the Atto488 signals IN: OUT of the FG particles. The FG particles were photobleached at 150 mM NaCl as in Fig. 5b. **c** Signal recovery in the FRAP experiments shown in **b** normalized to 1 for a complete recovery of the signal and plotted against time post bleaching. Five replicates with different FG particles are shown. The translational diffusion coefficients derived from fitting the recovery (solid lines) to theoretical recovery curves (Methods) are shown next to the recovery curves as mean ± S.D.

physical roles of Phe residues are not totally clear. In one possible scenario, Phe residues preferentially interact with each other via π-π stacking or T-stacking interactions and these interactions interplay with hydrophobic interactions contributed by non-specific hydrophobic groups in the system. In another scenario, aromatic groups are just non-specific components of a hydrophobic core, among all other hydrophobic groups. To resolve this question, information on the distances between aromatic groups within the FG phase would be valuable. Moreover, molecular dynamics (MD) simulations could also be useful in obtaining a complete picture of motion in the FG phase, including protein sidechains, and NMR data can then be used to calibrate and validate MD studies.

While FG motif-NTR interactions have been well-studied in solution[35,36,88], these interactions might show unprecedented

qualities when occuring within a condensed FG phase, as here we have shown that cohesive interactions and phase separation have an impact on the local FG dynamics. Here we demonstrated at the nanoscopic scale that prf.GLFG$_{52 \times 12}$ recapitulates many of the biophysical properties of native Nup98 FG domains, and is thus an ideal starting point for studying FG motif-NTR interactions. The methodology of applying NMR spectroscopy to measure the atomic-scale biophysical properties of an FG phase, and a perfectly repetitive sequence to simplify NMR analysis could also be extended to the study of interactions between FG domains and NTRs in the future.

Our study provides an example of entropy-driven phase separation. In contrast, enthalpy-driven phase separation shows an upper critical solution temperature (UCST) behaviour (i.e.,

higher temperature disfavours the phase separation), e.g., in case of IDPs enriched in both aromatic and charged residues, like DDX4[38] or hnRNPA1[37]. Given the lack of charged residues, in particular Arg, it is not surprising that we observed a different phase separation behaviour in Nup98 FG domains. It has been demonstrated that the UCST/LCST behaviours of resilin/elastin-like repeat proteins are dependent on the number of charged residues[89]. It is noteworthy that some other FG domains in NPCs, like those in Nsp1/Nup214/Nup153 also contain a significant percentage of charged residues[31] and some are less cohesive. It would be interesting to test the temperature dependence of these FG domains. From the perspective of eukaryotic biology, it is undesirable for the NPC barrier to be temperature sensitive, thus the temperature dependence of phase separation of Nup98 FG domains could have been counterbalanced by combining with other FG domains, including the less cohesive ones. Alternatively, the temperature dependence of phase separation could have been compensated through the grafting of the FG domains in the NPC scaffold at a high density.

## Methods

**DNA sequences of FG domain variants**. DNA sequences encoding the FG domain variants were generated with the assistance of the *Gene Designer* programme, which minimized the repetition of local DNA sequences and optimized codon usage for *E. coli* expression. DNA fragments were synthesized by *GenScript* and cloned into a bacterial expression vector for overexpression and purification (see below). See Supplementary Table 1 for a complete list of the plasmids and the sources.

### Recombinant protein expression and purification

*TtMacNup98A (Mac98A) FG domain*. Procedures were as published previously[27] with modifications here for $^{15}N/^{13}C$ isotope labelling: *E. coli* NEB Express cells transformed with a plasmid, encoding the protein fused to an N-terminal histidine-tag (Supplementary Note 1; Supplementary Table 1), were allowed to grow in a modified M9 medium (supplemented with trace elements) containing the isotopes (Supplementary Table 2) at 37 °C until OD$_{600}$ reached ~0.8. Expression of the target protein was induced by 0.4 mM IPTG, 30 °C for 16 h. Then the protein was extracted, purified by Ni chromatography and lyophilized as previously described. Lyophilized powder was weighed for quantification and stored at −20 °C until use.

*prf.GLFG$_{52 \times 12}$*. Since a histidine-tag may introduce unwanted signals in high-resolution NMR measurements, we designed a construct with a SUMO protease cleavage site[68,90] such that the histidine-tag was cleaved after Ni chromatography (as described below), namely His$_{14}$-ZZ-*sc*SUMO-prf.GLFG$_{52 \times 12}$ (plasmid number: 'pSNG064', see also Supplementary Table 1). The construct was expressed in *E. coli* cells in the M9 medium as described above. The cells were resuspended in cold 50 mM Tris/HCl pH 7.5, 300 mM NaCl, 300 mM (NH$_4$)$_2$SO$_4$, 1 mg/ml lysozyme, and lysed by a freeze-thaw cycle followed by a mild sonication. Inclusion bodies containing the construct were recovered by centrifugation (k-factor: 798; 45 min; 25 °C) and washed. The construct was extracted with 4 M guanidine hydrochloride, 50 mM Tris/HCl pH 7.5, 10 mM DTT. The extract was cleared by ultra-centrifugation (k-factor: 135; 90 min) and applied overnight at room temperature to a Ni(II) chelate column. The column was washed in extraction buffer containing 20 mM imidazole, then extensively in 0 M guanidine hydrochloride, 50 mM Tris/HCl pH 7.5, 10 mM DTT, and the proteins were eluted with 500 mM imidazole, 50 mM Tris/HCl pH 7.5, 10 mM DTT. The eluate was digested with 30 nM *bd*SENP1 protease[68] for 18 h, at 21 °C. The removal of the His$_{14}$-ZZ-*sc*SUMO tag from prf.GLFG$_{52 \times 12}$ decreased its solubility and induced its phase separation. Thus, the untagged prf.GLFG$_{52 \times 12}$, in form of insoluble material, was recovered by centrifugation (k-factor: 798, 25 °C, 1 h), and then washed twice in 300 mM ammonium acetate pH 7.5. The protein was solubilized with 30% acetonitrile. Residual uncut construct and His$_{14}$-ZZ-*sc*SUMO tag in the extract were cleared by passing the solution through a Ni(II) chelate column. The flow-through was lyophilized. The lyophilized powder was weighed for quantification and stored at −20 °C until use. Note that the product is completely free of cysteine residues and thus free of disulfide cross-links, which could change the local mobility of the peptide chain in the proximity of the cross-links.

For FRAP experiments, the FG domain was expressed with an additional C-terminal cysteine (plasmid number: pSNG102; Supplementary Table 1) for labelling with Atto488 maleimide. The domain was produced as described above, except that the bacteria were cultured in Terrific Broth (TB). The lyophilized, purified protein was dissolved in 2 M guanidine hydrochloride and the protein was allowed to react with Atto488 maleimide (ATTO-TEC, Germany) at a 1:1 molar ratio at pH 6.8. The labelled protein was further purified by gel filtration on a PD10

Sephadex column (GE Healthcare) and quantified by the absorbance of Atto488. Labelling was essentially complete, as assessed by SDS-PAGE.

*Pro-free prf.GLFG$_{52 \times 12}$*. Procedures were as described[27], except that M9 medium containing isotopes was used for cell culture.

*GLFG$_{52 \times 12}$, GAFG$_{52 \times 12}$, GXFG//L$_{52 \times 12}$, GXFG//V$_{52 \times 12}$ and GLLG//L$_{52 \times 12}$*. Procedures were the same as that for the Mac98A FG domain (for constructs containing a His$_{18}$-tag, see Supplementary Table 1)/prf.GLFG$_{52 \times 12}$ (for constructs containing a His$_{14}$-ZZ-*sc*SUMO tag), except that TB medium was used for cell cultures.

*prf.GLFG$_{7 \times 12}$ fragment (7x perfect repeats), GLLG$_{52 \times 12}$ and GLAG$_{52 \times 12}$*. His$_{14}$-ZZ-*sc*SUMO constructs similar to the above were constructed and expressed as described (the prf.GLFG$_{7 \times 12}$ fragment was expressed in the M9 medium containing $^{15}N$ isotope; GLLG$_{52 \times 12}$ and GLAG$_{52 \times 12}$ were expressed in TB medium). These constructs did not form inclusion bodies in the *E.coli* host and thus the soluble cell lysates cleared by ultracentrifugation were applied directly to Ni(II) chelate columns. The columns were washed extensively in 50 mM Tris/HCl pH 7.5, 300 mM NaCl, 20 mM imidazole, 20 mM DTT, and then in protease buffer: 50 mM Tris/HCl pH 7.5, 150 mM NaCl, 5 mM DTT. 50 nM *bd*SENP1 in protease buffer was applied for overnight on-column cleavage[68,90]. The cleaved target proteins were soluble and eluted with the protease buffer.

*Protein probes for FG particles permeation assays*. NTF2 and mCherry were expressed as His-tagged-fusions (Supplementary Table 1) and purified by native Ni(II) chelate chromatography, as described previously[26,29].

### FG particles preparation for permeation assays and confocal laser scanning microscopy

A previously described procedure[27] was applied with minor modifications: 2 µl of a 1 mM (~60 µg/µl) FG domain stock solution (for each, lyophilized powder dissolved in 2 M guanidine hydrochloride) was rapidly diluted with 100 µl (for Mac98A FG domain) or 50 µl (for prf.GLFG$_{52 \times 12}$) assay buffer (20 mM sodium phosphate buffer (NaPi) at pH 6.8, containing 150 mM NaCl, 5 mM DTT). For each, 15 µl of the mixture was mixed with 15 µl permeation probes containing 2 µM Alexa488-labelled NTF2 and 10 µM mCherry in assay buffer. Therefore, final concentrations of Mac98A FG domain/ prf.GLFG$_{52 \times 12}$ were 10 µM/ 20 µM, respectively. A lower assay concentration for Mac98A FG domain was used due to its lower critical concentration[27].The resulting mixture was placed on collagen-coated micro-slides 18-well (IBIDI, Germany). FG particles were allowed to sediment for 60 min to the bottom of the slide.

Then Alexa488 (for tracking NTF2) or mCherry signals were acquired with 488 or 561 nm excitation, respectively with a Leica SP5 confocal scanning microscope equipped with a 63x oil immersion objective and hybrid detectors (standard mode, in which nonlinear response of the detector was auto-corrected) at 21 °C. Partition coefficients (taken as signal inside FG phase: signal in buffer) were quantified as previously described[27]. Leica Application Suite X 3.3.0 was used for data acquisition and quantification.

### Fluorescence recovery after photobleaching (FRAP) and estimation of diffusion coefficients

For all, 2 µl of unlabelled FG domain stock (1 mM FG domain in 2 M guanidine hydrochloride) was rapidly diluted with 100 µl pre-cooled assay buffer (20 mM NaPi pH 6.8, 150 mM NaCl, 5 mM DTT or [NaCl] specified in the figures) on ice. At this point phase separation was suppressed by the ice temperature. Then 30 µl of the FG solution was quickly mixed with 1 µl of a given Atto488-labelled FG domain (Supplementary Table 1), which had been adjusted to a concentration of 15 µM, except for GAFG$_{52 \times 12}$: 90 µM. Therefore, the final protein concentration of unlabelled FG protein was 20 µM; the concentration of Atto488-labelled species was 0.5 µM (Atto488-labelled GAFG$_{52 \times 12}$: 3 µM). The mixtures were incubated at room temperature to allow phase separation and then transferred to micro-slides for confocal laser scanning microscopy at 21 °C. FG particles (*n* = 5 typically) of radii ≈ 4–5 µm were photobleached near the centres at 488 nm. The bleach area was manually defined to a circular region with a radius of 1.5 µm, however, we found that the actual bleached area was always ≈ 2–3 µm in radius, due to the diffusion during bleaching and Gaussian blurring of the laser beam[91]. After bleaching, Atto488 signals were acquired at 2 s intervals. Raw data were corrected for photobleaching during acquisition and normalized to the initial signal. The recovery curves were fitted to a simple exponential model[92] as follows:

$$C(t) = A\left(1 - \exp - (t/\tau)\right) \qquad (1)$$

where $C(t)$ is the normalized signal at time $t$, A is a preexponential scaling factor close to 1, which was used to account for incomplete recovery (due to a small immobile fraction of protein in the sample), and $\tau$ is the characteristic diffusion time, during which 1/e of the fluorescence recovers. Half-time for recovery ($t_{1/2}$) is

derived from $\tau$ by:

$$t_{1/2} = -ln0.5 \cdot \tau \qquad (2)$$

The diffusion coefficient ($D$) was then calculated from the half-time using the following equation[93]:

$$D = 0.224 \cdot R^2 / t_{1/2} \qquad (3)$$

where $R$ is the measured radius of the bleached spot (taken as the half-width at 86% of bleach depth[91]). Leica Application Suite X 3.3.0 was used for data acquisition and quantification. Microsoft Excel version 16.42 was used for data fitting (by the least-squares method) and calculations.

**Analysis of phase separation by centrifugation (centrifugation assay).** In general, each 1 µl of FG domain stock (1 mM protein in 2 M guanidine hydro-chloride) containing ~60 µg protein was rapidly diluted with assay buffer (typically 20 mM NaPi at pH 6.8, 5 mM DTT, 150 mM NaCl or [NaCl] specified in the figures) to the concentration stated in the figures. After 1 min, the FG phase (insoluble content) was pelleted by centrifugation (21130 g, 30 min, using a temperature-controlled Eppendorf 5424 R centrifuge equipped with an FA-45-24-11 rotor) at the temperature specified in the figures. The equivalent ratio (6%) of the pellet (FG phase) and supernatant was analysed by SDS-PAGE/ Coomassie blue staining. Critical concentration for phase separation of a given FG domain (or variant) was taken as the concentration that remained in the supernatant, which was estimated with a concentration series loaded onto the same gel. All experiments were repeated three times with similar results, and the representative gel images/ critical concentrations are shown. Full scans of gels with molecular weight markers are provided in the Source Data file. In those cases where the critical concentration is above 20 µM, the experiment was repeated by dissolving 600 µg of lyophilized powder of FG domain directly in assay buffer to a concentration of 100 µM, and the load amount on the gel was adjusted to 0.6% accordingly by pre-dilution.

**Analysis of phase separation by Dynamic Light Scattering (DLS).** In general, a stock solution of prf.GLFG$_{52 \times 12}$ was rapidly diluted with filtered NaCl solutions containing 20 mM NaPi pH 6.8, as described above, to protein concentrations and NaCl concentrations stated in Fig. 1f. For protein concentrations > 50 µM, lyophilized powder of prf.GLFG$_{52 \times 12}$ was dissolved and serially diluted in the NaCl solutions. 10 µl of each solution was analysed in a closed cuvette using a DynaPro NanoStar DLS instrument (Wyatt Technologies). To acquire temperature-dependent phase separation curves, the temperature was automatically raised by 1 °C per min, typically in the range of 2–40 °C (or as indicated in Fig. 1e, also cooling from 40 to 2 °C). DLS signal was acquired continuously. Phase transition was indicated by a sharp increase in the light scattering intensity, and the transition temperature was rounded up to the nearest 0.1 °C. The Wyatt Technologies Dynamics 7.1.5 software was used for autocorrelation analysis and computation of hydrodynamic radii. Three datasets for each condition were averaged, and standard deviations (S.D.) were shown as error bars in Fig. 1f.

### NMR measurements

*Sample packing.* Formation of FG phases has been observed by hydration of lyophilized protein powder[11,19]. For solid-state NMR measurements, we used a similar method: roughly 1.1 mg of lyophilized powder of a given FG domain was packed into 1.3 mm rotors. Using excess buffer, samples were hydrated in a tabletop centrifuge (Eppendorf 5418 R, rotor: FA-45-18-11) at 16873 g for 1 h at 25 °C. The buffer used for NMR spectroscopy contained 20 mM NaPi pH 6.8, 0.01% NaN$_3$, 5 mM EDTA and 150 mM NaCl. For tracking the effect of salt concentration, a series of buffers containing 75 mM, 200 mM, 300 mM, or 600 mM NaCl was applied.

Forming homogeneous gels in solution NMR tubes would have been advantageous, as it would have allowed the use of standard solution pulse sequences. This, however, was not possible, as the hydrogels exhibited severe magnetic susceptibility inhomogeneities, which could only be averaged out by magic angle spinning.

*General spectral parameters for solid-state measurements.* All spectra, except when noted otherwise were recorded in a 1.3 mm MAS probe at 20 kHz MAS spinning frequency in a 600 MHz wide bore Bruker Avance III HD spectrometer. Unless indicated otherwise, the temperature was set to 37 °C at 500 l/h VT gas flow. Typical pulse lengths were 2.5 µs on $^1$H, 4.2 µs on $^{13}$C, and 5.0 µs on $^{15}$N. In all experiments, where applicable, water suppression was achieved by a 100 ms MISSISSIPPI pulse train[94]. Decoupling during acquisition periods was set to 5 kHz TPPM[95] on protons, and to 5 kHz WALTZ[96] on $^{15}$N, and for double-labelled samples, also on $^{13}$C. Recycle delays were set to 1.0 s unless noted otherwise. Unless noted otherwise, all spectra were processed using a cosine squared window function using the Bruker Topspin 4.0.8. software package.

*MAS NMR $^{13}$C spectra with direct excitation and $^1$H-$^{13}$C cross-polarization.* $^{13}$C-detected experiments were recorded on u-[$^{13}$C-$^{15}$N] labelled samples, averaging 1024 scans. Cross-polarization (CP) spectra on FG phases assembled from *Tt*MacNup98A FG domain and prf.GLFG$_{52 \times 12}$ were recorded at an optimal CP

condition of $\omega_H^1 = \omega_C^{13}$, (average fields of 30 kHz on both nuclei), using an 80–100% ramp on $^{13}$C with an optimal length of 13 ms. For Pro-free prf.GLFG$_{52 \times 12}$, the optimal CP condition was found to be 50 kHz on $^{13}$C, using a 80–100% ramp, and 68 kHz on $^1$H (average fields), the length of the CP transfer was set to 4 ms. Relaxation delays were optimized to allow for full signal recovery between scans. Spectra were processed with an exponential window function, applying a line broadening corresponding to half the linewidth (60 Hz, 60 Hz, and 80 Hz, in Fig. 2b, top, middle and bottom, respectively).

*MAS NMR $^{15}$N-$^1$H and $^{13}$C-$^1$H correlation spectra.* $^{15}$N-$^1$H HSQC$^{SSMAS}$ (using refocused INEPT blocks to allow for MISSISSIPPI water suppression) and TROSY[97] spectra were recorded at 20 kHz MAS spinning frequency at 31 °C in a 600 MHz wide bore Bruker Avance III HD spectrometer. $^{15}$N-$^1$H INEPT delays were set to 1.8 ms. Each spectrum averaged four scans per point. Spectra were processed with 100 ms acquisition time in the indirect $^{15}$N, and 400 ms in the direct $^1$H dimension.

$^{13}$C-$^1$H HSQC$^{SSMAS}$ spectra were recorded with a pulse sequence analogous to the $^{15}$N-$^1$H HSQC$^{SSMAS}$ experiment. The $^{13}$C-$^1$H HSQC$^{SSMAS}$ experiment in Fig. 3c was recorded on a u-[$^{15}$N]-labelled sample without $^{13}$C-labelling in order to eliminate the effect of one-bond $^{13}$C-$^{13}$C scalar couplings to gain resolution. $^{13}$C-$^1$H INEPT delays were set to 1.1 ms. The $^{13}$C dimension was sampled to 40 ms.

The spectrum used for overlay with a solution spectrum (Supplementary Fig. 5) was measured on a u-[$^{13}$C,$^{15}$N]-labelled sample to make it comparable to the solution spectrum that had to be recorded on an isotopically labelled sample for sensitivity reasons. The cooling gas was set to a temperature of 23 °C at 500 l/h gas flow, which was found to be 31 °C at the sample after calibration. The spectrum was sampled to 50 ms in the $^{13}$C dimension, but only 10 ms were used for processing.

For the H(H)CH NOESY experiment the NOE mixing block was added to the $^{13}$C-$^1$H HSQC$^{SSMAS}$ block analogously as previously described[98]. The spectrum was recorded on an 800 MHz Bruker Avance III HD spectrometer in a narrow-bore 1.3 mm probe. All experimental conditions were identical as described above. The NOE mixing time was set to 100 ms. Assignment of the side chains was accomplished with a similar H(H)CH pulse sequence on the 600 MHz Avance III HD spectrometer, the $^1$H-$^1$H mixing correlating nuclei within the same spin system was achieved with a WALTZ-16[96] scheme with 12.5 µs pulse lengths applied for 9.6 ms.

*MAS NMR assignment experiments.* Assignment experiments were recorded on a Bruker Avance III HD 850 MHz spectrometer using a wide bore H/X/Y/D 3.2 mm probe. MAS spinning frequency was set to 12.5 kHz, the temperature of the cooling gas was set to 310 K. The sample was u-[$^{13}$C,$^{15}$N]-labelled. INEPT delays were set to 2.7 ms for the H/N transfer, 10 ms for the N/C, 4.5 ms for the CO-CA transfer, and 11.2 ms for the CB/CA and CA/N simultaneous transfer. The HNCA spectrum[99] was acquired in 20 h and was sampled to 18.6 ms in $^{15}$N, 24.8 ms in $^{13}$C, and 27.0 ms in the $^1$H dimension. The HN(CO)CA experiment[99] was acquired using the same sampling as the HNCA experiment. For the HNCACB experiment[99], acquired in 32 h, sampling was identical to the previous experiments in the $^1$H and $^{15}$N dimensions, and 6.0 ms in the $^{13}$C dimension. Although these acquisition times in the direct and indirect dimensions are not sufficient to take full advantage of the narrow linewidths offered by the sample, they provide sufficient resolution to accomplish the sequential assignment of the repeating units.

*Determining rotational correlation times with TRACT (MAS NMR).* TRACT experiments[79] were recorded using buffers with varying NaCl concentrations (75, 150, 200, 300, or 600 mM NaCl) and at two different temperatures (24 °C and 36 °C) at a NaCl concentration of 150 mM. Residue-specific rotational correlation times ($\tau_c$) were measured for the peptide chain using pseudo-3D experiments[79], where a series of 2D $^{15}$N-$^1$H planes were recorded with varying increments for both the $\alpha$ and the $\beta$ states. The $^1$H dimension was sampled to 100 ms, the $^{15}$N dimension to 40 ms (sweep widths were 41.7 ppm on $^1$H and 25.0 ppm on $^{15}$N). Each spectrum averaged 32 scans per point. For both states, seven points were sampled in the pseudo dimension such that signal in the last increment was ~30% of the original. Each peak in a pseudo-3D spectrum was fitted with an exponential decay using the Bruker Dynamics Center. Errors of the relaxation rates were estimated from the goodness of the fit. $\tau_c$-values were calculated as described by Lee et al.[79]. Errors of $\tau_c$ were determined from the errors of the relaxation rates using the Monte Carlo method (10000 runs). Experiments at varying NaCl concentrations were repeated on two independent samples with similar results.

*Temperature calibration.* Temperature calibration in solid-state probes was achieved using a solution consisting of 80% ethylene glycol and 20% DMSO-$d_6$ filled into a 1.3 mm rotor, which was then spun at 20 kHz and cooled/heated with the variable temperature gas flow. The exact temperature was determined from the chemical shift difference of the ethylene and the hydroxyl protons.

*Solution NMR experiments.* Solution samples ('soluble state') of prf.GLFG$_{52 \times 12}$ were prepared by dissolving the lyophilized sample (u-[$^{15}$N]-labelled for [$^{15}$N, $^1$H]-TRACT experiments, u-[$^{13}$C, $^{15}$N]-labelled for $^{15}$N-$^1$H and $^{13}$C-$^1$H HSQC spectra) in ice-cold buffer (20 mM NaPi, pH 6.8, 150 mM NaCl, unless otherwise specified) such that its

concentration was 15 µM. This solution was then centrifuged at 25 °C in a temperature-controlled tabletop centrifuge at 16873 g for 1 h. The supernatant (containing ~5 µM prf.GLFG$_{52 \times 12}$) was then quickly transferred to a 3 mm NMR tube. This procedure ensured the absence of phase separation in solution samples, since the critical concentration of prf.GLFG$_{52 \times 12}$ during centrifugation is lower than that at the temperature applied (24 °C) during the measurements. Since D$_2$O is necessary for locking, however, it could influence phase separation behaviour of FG domains, the 3 mm NMR tube was placed in a 5 mm tube containing pure D$_2$O. Due to the concentric placement of the tubes, the samples could still be shimmed adequately.

HSQC spectra were recorded on a 700 MHz Bruker Avance III HD spectrometer equipped with a 5 mm CPPTCI H/F-C/N/D Z-GRD cryoprobe. HSQC spectra were sampled to 1 s on $^1$H (for processing, only 400 ms were used) and 50 ms on the heteronucleus. The $^{15}$N-$^1$H HSQC spectra were recorded with the fhsqcf3gpph pulse sequence from the Bruker library, using the standard parameter set, the $^{13}$C-$^1$H HSQC spectra were measured with the hsqcetgpsi standard Bruker pulse sequence, also using the standard parameter set.

[$^{15}$N, $^1$H]-TRACT experiments at 0 mM salt concentration were measured on a 600 MHz Bruker Avance Neo spectrometer equipped with a 5 mm PH QXI-H/P-C/N-D-05 cryoprobe, at 150 mM NaCl conctration, TRACT was measured on a 700 MHz Bruker Avance III HD spectrometer equipped with a 5 mm CPPTCI H/F-C/N/D Z-GRD cryoprobe. All experiments were sampled to 20 ms on $^{15}$N (sweep width was 36 ppm), and 200 ms on $^1$H (sweep width was 13 ppm) averaging eight scans per point. 16 increments were measured for both states such that the signal in the last increment was ~30% of that in the first increment. $\tau_c$-values were obtained by solving the equation

$$\eta_{xy} = 2p\delta_N\left(3\cos^2\theta - 1\right)\left(4J(0) + 3J(\omega_N)\right) \qquad (4)$$

where $p = \frac{\mu_0\gamma_H\gamma_N h}{16\pi^2\sqrt{2}r_{NH}{}^3}$, $\delta_N = \gamma_N B_0\Delta\delta_N/(3\sqrt{2})$, and $J(\omega_N) = 0.4\tau_c/(1 + \omega^2\tau_c{}^2)$. $\gamma_H$ and $\gamma_N$ are the gyromagnetic ratios of protons, and $^{15}$N, respectively, $h$ is the Planck constant, $r_{NH}$ is the distance between the amide $^{15}$N and $^1$H, $B_0$ is the field of the NMR spectrometer, and $\tau_c$ is the rotational correlation time of the molecule or a selected residue.

The concentration of the prf.GLFG$_{7 \times 12}$ sample was 100 µM. All spectra at 24 °C and 150 mM NaCl concentration ([$^{15}$N, $^1$H]-TRACT, $^{15}$N-$^1$H HSQC) were recorded as described above on a 700 MHz Bruker Avance III HD spectrometer equipped with a 5 mm CPPTCI Z-GRD cryoprobe. TRACT experiments at 36 °C were measured on 700 MHz Bruker Avance Neo spectrometer equipped with a 5 mm CPPTCI H-C/N/D Z-GRD cryoprobe. TRACT experiments at 0 mM NaCl concentration were measured on a 600 MHz Bruker Avance Neo spectrometer equipped with a 5 mm PH QXI-H/P-C/N-D-05 cryoprobe. All spectra were processed using a cosine squared window function using the Bruker Topspin 4.0.8. software package.

**Statistics and reproducibility**. No statistical method was used to predetermine sample size. No data were excluded from the analyses. The experiments were not randomized. The investigators were not blinded to allocation during experiments and outcome assessment. Experiments shown in Figs. 1a, 1d, 5a, 7a and Supplementary Figs 2a, 2b, 6, 7 were repeated three times on independent samples with similar results. Experiments shown in Figs. 1c, 1g, 2a and 5c were repeated twice on independent samples with similar results.

**Reporting summary**. Further information on research design is available in the Nature Research Reporting Summary linked to this article.

## Data availability
A source data file is provided with this article. The data that support the findings of this study are provided in the Supplementary Information or Source Data file or are available from the corresponding authors upon reasonable request. Source data are provided with this paper.

## Code availability
The Jupyter notebook for fitting TRACT data is provided in Supplementary File 1.

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

## Acknowledgements

The authors thank Yunior Cabrales Fontela and Rıza Dervişoğlu for their help with preliminary NMR measurements, as well as the Max-Planck-Gesellschaft and the Deutsche Forschungsgemeinschaft for funding (SFB 860 Project B11 to L.B.A and D.G.).

## Author contributions

E.E.N. and S.C.N. planned and conducted all the experiments. C.G., D.G. and L.B.A. conceived the overall concepts of the study. All authors contributed to experiment design, data analysis, interpretation, and manuscript writing.

## Funding

## Competing interests

The authors declare no competing interests.
