## [Peer Review File · Nature Communications]

Atomic resolution dynamics of cohesive interactions in phase-separated Nup98 FG domainsREVIEWERS' COMMENTS

Reviewer #1 (Remarks to the Author):

The work reveals the atomic level information on the dynamics of cohesive interactions in both phase-separated and soluble states of engineered FG domains using solution and magic angle-spinning NMR. This interesting approach allowed the measurement of residue-specific local dynamics to understand the dynamics of inter-FG repeat cohesive interactions within the FG phase. The authors could directly detect contacts involved in cohesive interactions. The FG domains of various homologues encode ~40-50 imperfect repeats presenting a technical obstacle for high resolution NMR due to the substantial overlap of resonances and the presence of many similar sub-sequences. The authors took the opposite route and utilized a perfect repeat (prf.GLFG52x12) consisting of 52 connected 12 amino-acid peptides GGLFGGNTQPAT, which simplifies the system allowing residue-specific measurements.

In my view, this work is decent, and the methodology was conducted with care. The idea is interesting, the manuscript is written well, and the results were presented attractively. However, I think that the authors should address the following issues to improve the quality of this review article. Specific suggestions are made below for improvement, including additions that can enhance the effectiveness of the story.

1. The authors can briefly suggest how the work can be useful to study the interactions between phase-separated FG domains and nuclear transport receptors.
2. The biophysical roles of cohesive interactions in the liquid FG-Nups, with respect to condensate formation and regulation, material state and reversibility, selective permeability, and structure-function relationships, may be discussed.
3. The sections are divided into several small paragraphs, a few of them just 4-5 lines. They can be merged into larger paragraphs.

Reviewer #2 (Remarks to the Author):

This is a very important study on the detailed interaction of FG-repeats within the nuclear pore complex.

FG-repeats are distinct elements within disordered regions of a set of nucleoporins that are responsible for generating the selectivity filter of the NPC. An age-old question is how selectivity and very high transport rates are accomplished at the same time.

Here, the authors provide new insight into the molecular mechanism of how the selectivity works. This is achieved by studying a dodecameric FG-peptide, with 52 perfect repeats, thus making it available for detailed, residue-specific NMR analysis. The authors establish that cohesive FG-repeats exhibit fast local mobility while maintain slow translational mobility. This behavior very likely explains how FG-interacting proteins can quickly 'melt' into FG-hydrogels as the entropic barrier is very small, as the authors conclude. This molecular understanding likely has far-reaching consequences for the entire field studying membrane-less condensates, of which the FG-hydrogel very much is the poster child.

The main results are elegantly controlled and tested under various temperature and salt conditions, in addition to extensive mutational analysis.

All in all, a flawless manuscript, expertly executed, and written with extraordinary clarity and authority. I am in favor of publication without revision.

Reviewer #3 (Remarks to the Author):

The work presents a very thorough characterization of cohesive interaction in a low complexity phase-separated FG domain example. Biophysical characterization is very carefully

conducted and care is taken to explain how each experiment demonstrates the results. This work is important for the field because it provides one of the few attempts of the mechanistic characterization of these interactions. The measurements of dynamics using NMR is a crucial component. For qualitative assessment an assumption of brownian motion is used even in the phase-separated state and is appropriate to demonstrate the importance of the hydrophobic residues.

My only suggestion in this regard is to consider an alternative interpretation of the data in which instead of increase in τ_c (Figure 4 b, phase separated state), instead an extent of diffusion anisotropy in motion is estimated, if it is feasible. While it will not change the qualitative picture, it might offer additional mechanistic insights, in particular possibly into the mode of interaction for Phe residues mentioned in the outlook section pi-pi vs T-stacking. The complexity of the analysis may be outside the scope of the present work, however the author may mention that details of the relaxation data may be a venue for future mechanistic insights

Answers to the reviewers' comments

(for clarity, the authors repeat reviewers' points in blue in front of each reply)

Reviewer #1 (Remarks to the Author):

The work reveals the atomic level information on the dynamics of cohesive interactions in both phase-separated and soluble states of engineered FG domains using solution and magic angle-spinning NMR. This interesting approach allowed the measurement of residue-specific local dynamics to understand the dynamics of inter-FG repeat cohesive interactions within the FG phase. The authors could directly detect contacts involved in cohesive interactions. The FG domains of various homologues encode ~40-50 imperfect repeats presenting a technical obstacle for high resolution NMR due to the substantial overlap of resonances and the presence of many similar sub-sequences. The authors took the opposite route and utilized a perfect repeat (prf.GLFG_{52x12}) consisting of 52 connected 12 amino-acid peptides GGLFGGNTQPAT, which simplifies the system allowing residue-specific measurements. In my view, this work is decent, and the methodology was conducted with care. The idea is interesting, the manuscript is written well, and the results were presented attractively.

Thank you very much for this enthusiastic summary. The positive evaluation is very encouraging to us.

However, I think that the authors should address the following issues to improve the quality of this review article. Specific suggestions are made below for improvement, including additions that can enhance the effectiveness of the story.

1. The authors can briefly suggest how the work can be useful to study the interactions between phase-separated FG domains and nuclear transport receptors.

The interactions between phase-separated FG domains and nuclear transport receptors (NTRs) remain challenging to study. The reasons for this are that:

- i) Prototypic NTRs are large molecules (e.g., Importin β : 876 aa; exportin CRM1: 1071 aa) that are not easily analysed by NMR.
- ii) contacts between FG motifs and NTRs are expected to be weak and transient, and are thus difficult to detect directly. In addition, within an FG phase, signals of these contacts may be masked by signals of FG-FG interactions.
- iii) There are multiple FG interaction sites on a given NTR. For example, although crystal structures showed that Importin β has a structurally well-defined binding pocket for FG motif around Ile178 (Bayliss et al., 2000), mutational analysis and molecular dynamics simulations suggested additional FG interaction sites (Kutay et al., 1997; Bednenko et al., 2003; Isgro and Schulten, 2005). A crystal structure of the exportin CRM1 revealed eight FG interaction sites, while site-directed mutagenesis indicated an even greater number (Port et al., 2015). Moreover, single amino acid residues like Arg, Cys and hydrophobic residues on NTR surfaces could also contact FG motifs (Frey et al., 2018). The overall heterogeneity of these multivalent contacts increases the complexity of the analyses.

However, our finding that many nanoscopic biophysical properties of prf.GLFG_{52x12} resemble those of native FG domains imply that prf.GLFG_{52x12} may be used as a model for studying interactions between phase-separated FG domains and NTRs. The reduced sequence

heterogeneity of prf.GLFG_{52x12} already reduces the complexity of the problem. We have added a sentence in the Outlook section to mention this point:

“Here we demonstrated at the nanoscopic scale that prf.GLFG_{52x12} recapitulates many of the biophysical properties of native Nup98 FG domains, and thus it is an ideal starting point for studying FG motif-NTR interactions. The methodology of applying NMR spectroscopy to measure the atomic-scale biophysical properties of an FG phase, and a perfectly repetitive sequence to simplify NMR analysis could also be extended to the study of interactions between FG domains and NTRs in the future.”

2. The biophysical roles of cohesive interactions in the liquid FG-Nups, with respect to condensate formation and regulation, material state and reversibility, selective permeability, and structure-function relationships, may be discussed.

We have modified a sentence in the Results section to mention the possible role of cohesive interactions in the viscosity of FG phase.

*“We previously showed that this difference (when compared to the Mac98A FG domain) is due to the lack of the 44-residue GLEBS domain (binding site for the mRNA export mediator Gle2p^{71,72}) in prf.GLFG_{52x12}²⁷, which also contributes to cohesive interactions, **and suggested that the overall strength of cohesion could affect the translational mobility of molecules and viscosity of the FG phase material.**”*

Moreover, we have also added a sentence to the Outlook section to discuss the role(s) of cohesive interactions in FG phase formation and permeation selectivity.

“Cohesive interactions are fundamental for the formation of such a barrier, concentrating and assembling the FG repeats into a condensed phase, while the permeation selectivity is observed because mobile species encounter the chemical properties of the FG phase, and beyond a certain size must overcome the self-affinity, i.e., cohesion, of the FG repeats.”

3. The sections are divided into several small paragraphs, a few of them just 4-5 lines. They can be merged into larger paragraphs.

We have merged several short paragraphs to make the text easier to follow.

Reviewer #2 (Remarks to the Author):

This is a very important study on the detailed interaction of FG-repeats within the nuclear pore complex. FG-repeats are distinct elements within disordered regions of a set of nucleoporins that are responsible for generating the selectivity filter of the NPC. An age-old question is how selectivity and very high transport rates are accomplished at the same time. Here, the authors provide new insight into the molecular mechanism of how the selectivity works. This is achieved by studying a dodecameric FG-peptide, with 52 perfect repeats, thus making it available for detailed, residue-specific NMR analysis. The authors establish that cohesive FG-repeats exhibit fast local mobility while maintain slow translational mobility. This behavior very likely explains how FG-interacting proteins can quickly ‘melt’ into FG-

hydrogels as the entropic barrier is very small, as the authors conclude. This molecular understanding likely has far-reaching consequences for the entire field studying membrane-less condensates, of which the FG-hydrogel very much is the poster child.

The main results are elegantly controlled and tested under various temperature and salt conditions, in addition to extensive mutational analysis.

All in all, a flawless manuscript, expertly executed, and written with extraordinary clarity and authority. I am in favor of publication without revision.

Thank you very much. These positive comments are very encouraging to us and are a true reward for our efforts.

Reviewer #3 (Remarks to the Author):

The work presents a very thorough characterization of cohesive interaction in a low complexity phase-separated FG domain example. Biophysical characterization is very carefully conducted and care is taken to explain how each experiment demonstrates the results. This work is important for the field because it provides one of the few attempts of the mechanistic characterization of these interactions. The measurements of dynamics using NMR is a crucial component. For qualitative assessment an assumption of brownian motion is used even in the phase-separated state and is appropriate to demonstrate the importance of the hydrophobic residues.

My only suggestion in this regard is to consider an alternative interpretation of the data in which instead of increase in τ_c (Figure 4 b, phase separated state), instead an extent of diffusion anisotropy in motion is estimated, if it is feasible. While it will not change the qualitative picture, it might offer additional mechanistic insights, in particular possibly into the mode of interaction for Phe residues mentioned in the outlook section pi-pi vs T-stacking. The complexity of the analysis may be outside the scope of the present work, however the author may mention that details of the relaxation data may be a venue for future mechanistic insights

We are thankful for this suggestion, indeed determining diffusion anisotropy from relaxation measurements is possible (demonstrated in e.g., Tjandra et al., 1995). These measurements might yield valuable insights into the Phe-Phe interactions, and would be an interesting avenue for future investigations. With TRACT we measured effective rotational correlation times, which could be fitted to various motional models, including anisotropic diffusion. This would require extensive relaxation measurements, and would ideally be analysed in the context of molecular dynamics simulations. We have now added:

“Note that these qualitative changes could be interpreted in the context of different motional models, including anisotropic rotational diffusion, however this would require extensive further relaxation measurements.”

Literature cited in this document:

- Bayliss, R., Littlewood, T., & Stewart, M. (2000). Structural Basis for the Interaction between FxFG Nucleoporin Repeats and Importin- β in Nuclear Trafficking. *Cell*, *102*(1), 99-108.
- Bednenko, J., Cingolani, G., & Gerace, L. (2003). Importin beta contains a COOH-terminal nucleoporin binding region important for nuclear transport. *J Cell Biol*, *162*(3), 391-401.
- Frey, S., Rees, R., Schünemann, J., Ng, S. C., Fünfgeld, K., Huyton, T. et al. (2018). Surface properties determining passage rates of proteins through nuclear pores. *Cell*, *174*(1), 202-217.e9.
- Isgro, T. A., & Schulten, K. (2005). Binding dynamics of isolated nucleoporin repeat regions to importin- β . *Structure*, *13*(12), 1869-1879.
- Kutay, U., Izaurralde, E., Bischoff, F. R., Mattaj, I. W., & Görlich, D. (1997). Dominant-negative mutants of importin-beta block multiple pathways of import and export through the nuclear pore complex. *EMBO J*, *16*(6), 1153-1163.
- Port, S. A., Monecke, T., Dickmanns, A., Spillner, C., Hofele, R., Urlaub, H. et al. (2015). Structural and Functional Characterization of CRM1-Nup214 Interactions Reveals Multiple FG-Binding Sites Involved in Nuclear Export. *Cell Rep*, *13*(4), 690-702.
- Tjandra, N., Feller, S. E., Pastor, R. W., & Bax, A. (1995). Rotational diffusion anisotropy of human ubiquitin from ^{15}N NMR relaxation. *Journal of the American Chemical Society*, *117*(50), 12562-12566.